# Multivalent cross-linking of actin filaments and microtubules through the microtubule-associated protein Tau

Yunior Cabrales Fontela[1,2,3], Harindranath Kadavath[1,3], Jacek Biernat[4], Dietmar Riedel[5], Eckhard Mandelkow[4,6] & Markus Zweckstetter [1,2,3]

Microtubule-associated proteins regulate microtubule dynamics, bundle actin filaments, and cross-link actin filaments with microtubules. In addition, aberrant interaction of the microtubule-associated protein Tau with filamentous actin is connected to synaptic impairment in Alzheimer's disease. Here we provide insight into the nature of interaction between Tau and actin filaments. We show that Tau uses several short helical segments to bind in a dynamic, multivalent process to the hydrophobic pocket between subdomains 1 and 3 of actin. Although a single Tau helix is sufficient to bind to filamentous actin, at least two, flexibly linked helices are required for actin bundling. In agreement with a structural model of Tau repeat sequences in complex with actin filaments, phosphorylation at serine 262 attenuates binding of Tau to filamentous actin. Taken together the data demonstrate that bundling of filamentous actin and cross-linking of the cellular cytoskeleton depend on the metamorphic and multivalent nature of microtubule-associated proteins.

[1] Deutsches Zentrum für Neurodegenerative Erkrankungen (DZNE), Von-Siebold-Strasse 3a, 37075 Göttingen, Germany. [2] Klinik für Neurologie, Universitätsmedizin Göttingen, Robert-Koch-Strasse 42, 37099 Göttingen, Germany. [3] Department of NMR-Based Structural Biology, Max-Planck-Institut für Biophysikalische Chemie, Am Faßberg 11, 37077 Göttingen, Germany. [4] Deutsches Zentrum für Neurodegenerative Erkrankungen (DZNE), Ludwig-Erhard-Allee 2, 53175 Bonn, Germany. [5] Max-Planck-Institut für Biophysikalische Chemie, Am Faßberg 11, 37077 Göttingen, Germany. [6] CAESAR Research Center, Ludwig-Erhard-Allee 2, 53175 Bonn, Germany. Correspondence and requests for materials should be addressed to M.Z. (email: Markus.Zweckstetter@dzne.de)

Microtubule-associated proteins (MAPs) bind to stabilize and promote assembly of microtubules[1, 2]. In addition, MAPs bundle actin filaments and cross-link the cellular cytoskeleton formed by microtubules and actin filaments (Fig. 1a)[3–10]. The interaction of MAPs with actin is important for neurite outgrowth[11, 12]. Representative MAPs are MAP1a, MAP1b, MAP2a, MAP2b, MAP2c, MAP4, and Tau, and isoforms of these proteins, which are often generated by alternative splicing[2]. Tau occurs in six different isoforms in the human central nervous system[13, 14]. The Tau isoforms differ in the number of N-terminal inserts and have either three or four imperfect repeats in their C-terminal half[15]. The imperfect repeats are important for binding to both microtubules and actin. In addition, short fragments of the microtubule-binding domain of Tau promote actin bundling[16]. Proteins from the MAP2 family also bind to actin filaments through their repeat domain[8]. Because a single repeat interacts with both monomeric and filamentous actin, but does not bundle actin filaments, more than one microtubule-binding repeat is believed to be required for bundling of filamentous actin (F-actin)[17]. Little is known, however, about the molecular nature of the Tau/F-actin complex, about the involved binding sites, the mechanism of F-actin bundling and the MAP-driven process of cross-linking of microtubules and actin filaments.

Misfolding and aberrant accumulation of Tau is a pathological hallmark of Alzheimer's disease (AD)[18, 19]. Occurrence of Tau deposits correlates with the loss of cognitive functions in AD, suggesting a connection between Tau and synaptic transmission[20]. Because Tau directly interacts with actin and regulate its stability[3, 4, 21, 22], the interaction of Tau with actin might influence synaptic plasticity. This is particularly the case for the interaction with filamentous actin, which plays an important role in dendritic spine morphology and postsynaptic reorganizations[23]. In agreement with the disease relevance of the Tau/actin interaction, actin-rich paracrystalline inclusions, so-called Hirano bodies, are found in brain histopathological samples of AD and related tauopathies[24, 25]. In addition, acetylation of Tau modulates actin polymerization and enhances synaptic dysfunction[26].

A molecular understanding of the interaction of MAPs with filamentous actin is complicated by the dynamic structure of microtubule-associated proteins[15, 27]. MAPs such as Tau belong to the class of intrinsically disordered proteins, which do not fold into a stable three-dimensional structure[28]. In addition, intrinsically disordered proteins often contain multiple binding sites with each binding site only providing low affinity for a target protein. In case of F-actin, the high molecular weight of actin filaments further complicates structural analysis of the complex between actin filaments and MAPs. Using a combination of NMR spectroscopy and other biophysical methods, we here dissect the regions in both Tau and F-actin that are important for complex formation. We then provide high-resolution information about the structural changes that occur when Tau binds to F-actin and show that conformational changes in specific regions, which are separated by flexible linkers, are responsible for bundling of actin filaments as well as for cross-linking of actin filaments with microtubules. Our results thus clarify the nature of interaction between actin and MAPs and the actin/microtubule crosstalk.

## Results

**Tau interacts with and bundles actin filaments**. The ability of Tau to bundle F-actin was studied using electron microscopy (EM). The diameter of single actin filaments was approximately 7 nm, in agreement with previous studies[29, 30]. Addition of an equimolar concentration of Tau caused bundling of F-actin (Fig. 1b), although some single filaments remained. Actin bundles ranged between 20 and 150 nm in diameter. Next, Tau/F-actin co-sedimentation was performed. The same sample, which was used for EM, was exposed to two sequential steps of centrifugation, in which single-actin filaments and actin bundles were collected separately. Subsequently, aliquots from supernatant (SN), filament (PF), and bundle (PB) fractions were loaded in a 4–20% gradient gel (Fig. 1c). Quantification of the intensity of different bands indicated that ~63% of Tau remained in the supernatant. In addition, ~8% of Tau molecules were bound to single filaments, while ~29% of Tau was found together with actin bundles. The data demonstrate that Tau bundles actin filaments.

In order to gain insight into the affinity of the Tau/F-actin-interaction, the change in fluorescence of 7-chloro-4-nitrobenz-2-oxa-1,3-diazole (NBD)-labeled F-actin upon addition of Tau was followed[31]. 0.2 μM of NBD-labeled and phalloidin-stabilized F-actin were mixed with increasing amounts of Tau (Fig. 1d and Supplementary Fig. 1). A fit to the experimental data using a single-binding site model resulted in a dissociation constant of 60 ± 10 nM. Other stoichiometries, however, cannot be excluded. The derived dissociation constant is lower than a previously reported value of 241 ± 43 nM[7]. This is likely due of differences in ionic strength, because previous studies used phosphate buffer[7], while we chose general actin assembly buffer.

In order to identify residues in Tau that bind to F-actin, we used NMR spectroscopy. F-actin and [15]N-labeled Tau were mixed in a 2:1 molar ratio and a two-dimensional (2D) [1]H-[15]N heteronuclear single-quantum coherence (HSQC) experiment was recorded. Comparison with a reference spectrum of Tau alone showed that addition of F-actin into the Tau pool decreased the signal intensity of several Tau residues (Fig. 1e and Supplementary Fig. 2). Very little chemical shift changes were observed, indicating that the binding process is intermediate to slow on the NMR time scale. According to the resonance assignment of Tau[27], cross-peaks from residues such as S262, N265, and T319 were strongly attenuated. Other residues including T175 and T217 were less affected, and S46 and S64 were not perturbed. Figure 1f highlights the sequence specific intensity ratio of all non-overlapping cross-peaks of Tau in the presence and absence of F-actin. High ($I_{bound}/I_{free}$) values are expected for residues that do not directly bind to F-actin. In contrast, residues, which are bound to F-actin, should have low ($I_{bound}/I_{free}$) values, because their resonances would—due to the high molecular weight of the complex—be attenuated. The analysis reveals that the proline-rich regions P1–P2 and the four pseudo-repeats contribute to the Tau/F-actin interaction. The strongest NMR signal attenuation was observed for K254-V287, K294-K331, and E342-D358 (Fig. 1f), suggesting that these Tau residues have the highest affinity toward F-actin. Consistent with this hypothesis, the cross-peaks of these residues were attenuated by F-actin also at higher ionic strength (Supplementary Fig. 3). Sequence analysis showed that K254-V287, K294-K331, and E342-D358 comprise some of the most hydrophobic parts of the Tau sequence (Supplementary Fig. 3a)[27].

**Tau binds to the hydrophobic pocket of F-actin**. In order to gain insight into how Tau regulates the polymerization and structure of F-actin, we sought to obtain information about the Tau-binding site on F-actin. X-ray crystallography and electron microscopy showed that several actin-binding proteins bind to a hydrophobic pocket between subdomains 1 and 3 of G-actin[32]. One of these binding partners is cofilin, a 21 kDa eukaryotic protein, which binds to F-actin with a $K_d < 0.05$ μM[33] and results in disassembly of F-actin[34, 35]. Cryo-electron microscopy further showed that the binding site of cofilin on F-actin is highly similar to its binding site on G-actin (Fig. 2a)[32]. To test if the binding site of Tau on F-actin overlaps with that of cofilin, we added a 10-fold

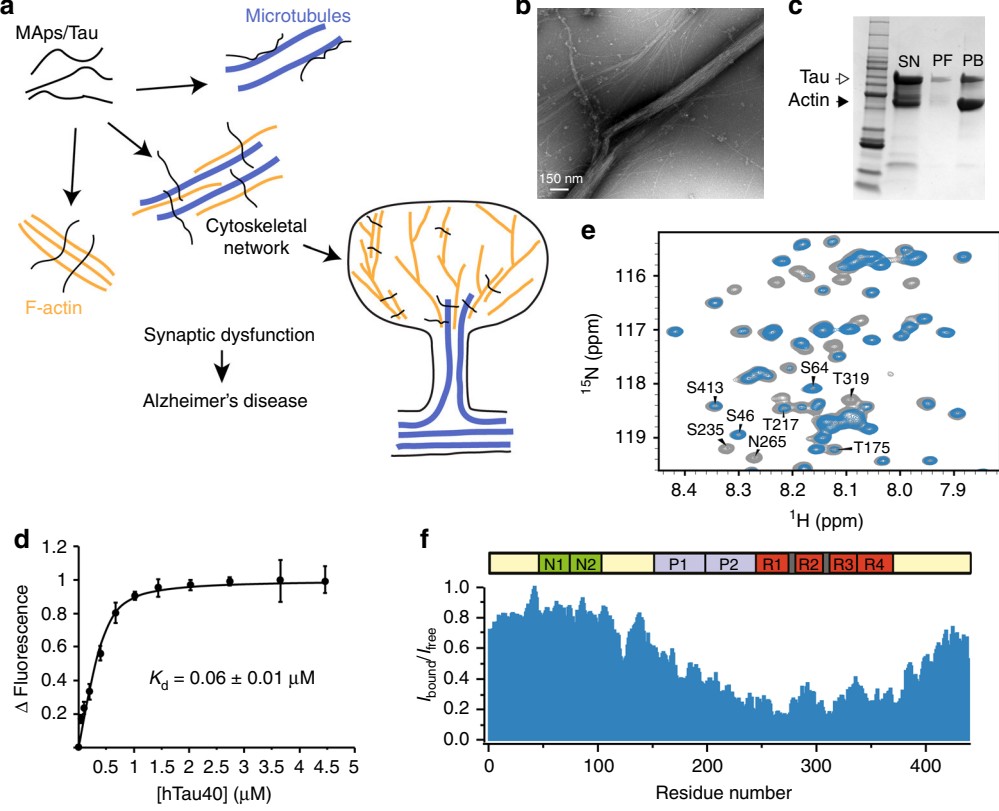

**Fig. 1** Tau interacts with and bundles filamentous actin. **a** Schematic representation of the importance of MAPs for the cellular cytoskeleton. MAPs (black) regulate microtubule dynamics (blue), bundle actin filaments (yellow), and cross-link actin filaments and microtubules. Aberrant interaction of Tau with F-actin is associated with synaptic dysfunction in Alzheimer's disease. **b** Electron micrograph of actin bundles induced by Tau. **c** Differential centrifugation in combination with a 4–20% gradient gel shows that Tau is associated with actin bundles. Lanes correspond to supernatant (SN), single filaments (PF), and bundles (PB). Open and filled arrowheads mark Tau and actin bands, respectively. **d** Affinity of the Tau/F-actin interaction measured by fluorescence using NBD-labeled F-actin. Error bars represent s.d. from three experiments. **e** Selected region of two-dimensional $^1$H-$^{15}$N HSQC spectra of 441-residue Tau in the absence (gray) and presence of a two-fold excess of F-actin (blue). **f** Residue-specific changes in $^1$H-$^{15}$N HSQC signal intensities of Tau upon addition of F-actin (shown in **e**). $I_{free}$ and $I_{bound}$ are intensities in the absence and presence of a two-fold excess of F-actin

excess of cofilin to a solution containing $^{15}$N-labeled Tau and 1.5-fold excess (with respect to Tau) of F-actin. For the resulting mixture, a $^1$H-$^{15}$N HSQC spectrum was recorded. After addition of cofilin, Tau's NMR resonance intensities and positions were similar to those in the absence of F-actin (Fig. 2b), indicating that (i) Tau is no longer bound to F-actin and (ii) cofilin does not bind to Tau. Subsequently, we repeated the actin/Tau co-sedimentation assay in the presence of cofilin. In agreement with the NMR data, addition of cofilin decreased the amount of Tau that co-sedimented with F-actin (Fig. 2c). The observation that cofilin, but no Tau, co-precipitates with actin filaments (Fig. 2c) also indicates that actin filaments were not disassembled by cofilin.

To further support binding of Tau to the hydrophobic pocket between subdomains 1 and 3 of actin, we used the effect of paramagnetic centers on NMR resonances. Particularly powerful are paramagnetic nitroxide tags such as (1-oxy-2,2,5,5-tetra-methyl-d-pyrroline-3-methyl)-methanethiosulfonate (MTSSL), because the attachment of MTSSL to a cysteine in the protein causes enhanced relaxation of nearby (less than ~25 Å from the paramagnetic center) protons[36]. Although actin contains five cysteine residues, a number of studies has shown that the C-terminal C374, which is in close proximity to actin's hydrophobic pocket (Fig. 2d), is most solvent exposed (Supplementary Fig. 4a, b) and incorporates more than 70% of spin labels during spin labeling reactions[37–41]. We thus labeled G-actin primarily at C374

with MTSSL, followed by polymerization of the MTSSL-tagged protein into F-actin. Subsequently, $^{15}$N-labeled Tau protein was added to reach a molar ratio of 1:2 (F-actin in excess) and a $^1$H-$^{15}$N HSQC of Tau was recorded. Comparison with the spectrum recorded in the presence of diamagnetic F-actin revealed residue-specific paramagnetic relaxation enhancement (PRE) (Fig. 2e). The PRE profile contained seven peaks corresponding to Tau residues L243-A246, K259-K267, V275-L284, S289-S293, S305-V313, S320-H330, and K375-F378 (Fig. 2e, Supplementary Fig. 4c).

Observation of PRE broadening at the chemical shift of unbound Tau demonstrates that the Tau molecule exchanges between the F-actin-bound and free state. Because of the simultaneous observation of line broadening upon addition of diamagnetic F-actin (Fig. 1f; Supplementary Fig. 2), the exchange process is intermediate on the NMR chemical shift time scale. In addition to residue-specific PRE effects within and close to the repeat domain, a gradually increasing PRE effect was observed from approximately residue 170 toward the N terminus. Because this region showed less sequence-specific variation, we suggest that the broadening effect is the result of unspecific binding of the N-terminal half of Tau to the surface of F-actin. In addition, there might be contributions from paramagnetic broadening, when individual actin filaments come together in bundles, as well as from cysteine residues other than C374, which might have been labeled by MTSL at low levels. In contrast to the gradually

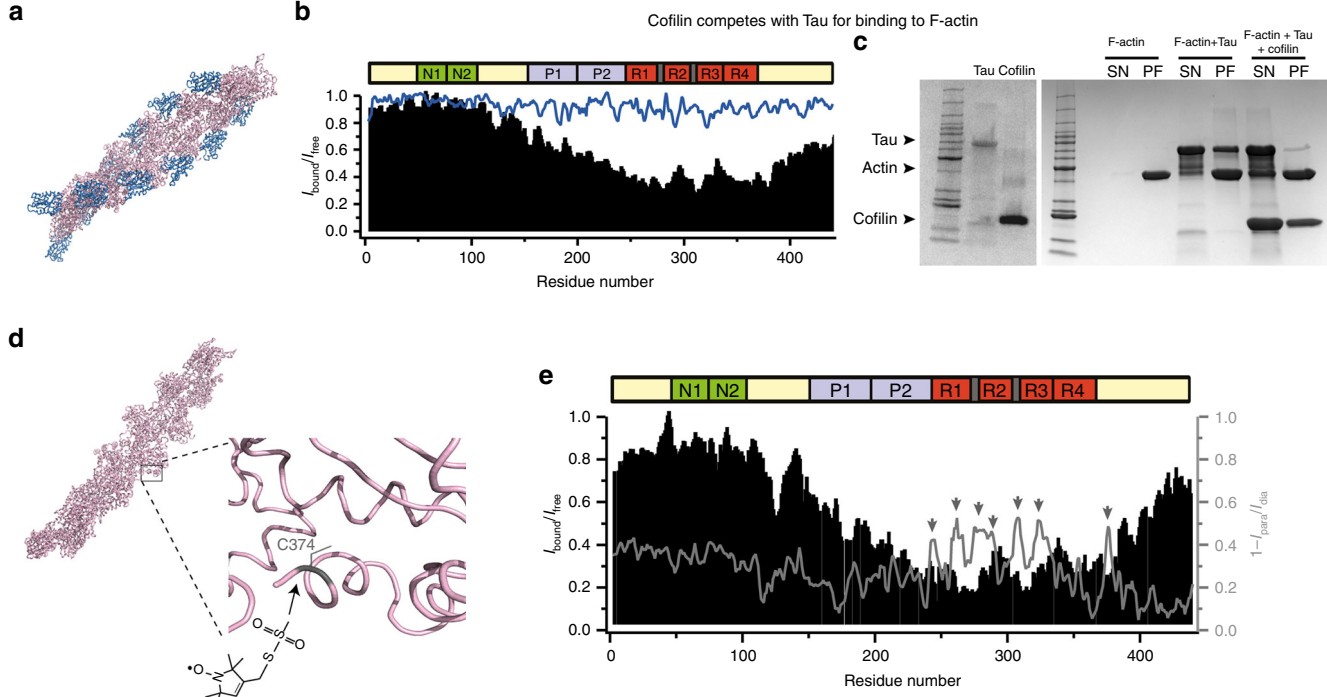

**Fig. 2** Tau binds to the hydrophobic pocket of actin. **a** 3D structure of F-actin (pink) in complex with cofilin (light blue) (PDB id: 3J0S). **b** Influence of a 10-fold excess of cofilin on $^1$H-$^{15}$N cross-peak intensities of Tau in presence of F-actin (Tau/F-actin molar ratio of 1:1.5). Normalized signal intensities in the absence (black) and presence of cofilin (blue) are shown. **c** Centrifugation-based sedimentation of F-actin, F-actin + Tau (1.5:1) and F-actin + Tau+cofilin (molar ratio 1.5:1:10). Supernatants (SN) and pellets (PF) were loaded in different lanes of a 4–20% SDS gradient gel. Arrowheads from top to the bottom represent Tau, actin and cofilin bands, respectively. **d** Cys374 of actin is solvent exposed and can be labeled with the spin label MTSSL[37–41]. **e** Sequence-specific paramagnetic broadening induced in Tau by MTSSL-tagged F-actin (gray line). $I_{para}$ and $I_{dia}$ are signal intensities observed for individual cross-peaks in two-dimensional $^1$H-$^{15}$N HSQCs of Tau in the presence of a two-fold excess of paramagnetic and diagmagnetic F-actin, respectively. For comparison, the attenuation of Tau signals by diamagnetic F-actin is shown (black bars)

increasing PRE effect in the N-terminal half of the protein, very little PRE effect was observed for residues 400 to 441 at the C terminus of Tau. The combined data—(i) competition of binding of Tau to F-actin by cofilin, which interacts with actin's hydrophobic pocket, and (ii) residue-specific PRE effects in the repeat domain of Tau by preferential MTSSL-labeling of C374 in proximity to the hydrophobic pocket—suggest that Tau binds to the solvent-exposed hydrophobic pocket that is located between subdomains 1 and 3 of actin.

**K18 bundles filamentous actin**. Truncation of Tau by caspases and endopeptidases has been suggested to constitute an important pathogenic step in AD[42, 43]. Many of the released fragments comprise part of the repeat domain of Tau and were found to form insoluble aggregates more rapidly[44]. Supported by these findings, several studies have used an artificial fragment of Tau, K18, which comprises all four repeats of the largest Tau isoform (residues Gln$^{244}$–Glu$^{372}$ plus initial Met$^{243}$) (Supplementary Fig. 5a), in order to gain insight into pathogenic processes in AD. We therefore tested whether K18 is able to bind to and bundle filamentous actin. Indeed, K18 bound with a Kd of 110 ± 1 nM to F-actin (Supplementary Fig. 1). The lower affinity when compared to full-length Tau is in agreement with the contribution of regions outside of the repeat domain to the F-actin interaction (Figs. 1f and 2b, e; Supplementary Fig. 2). In addition, K18 was capable of bundling F-actin (Supplementary Fig. 5b).

**Tau binds through helical structure to F-actin**. The above data suggest that Tau does not bind as a single rigid entity to F-actin.

Instead, short hydrophobic Tau segments, which are separated from each other by flexible residues, bind dynamically to actin's hydrophobic pocket (Fig. 2). To gain insight into the structure, which these segments adopt in complex with F-actin, we used the peptide Tau(254–290), which bundles filaments (Fig. 3a). NMR Nuclear overhauser effect (NOE) measurements that are sensitive for three-dimensional structure were subsequently performed. Because the peptide cannot be directly observed by solution-state NMR in complex with the megadalton-sized F-actin, we took advantage of the lower affinity and high off-rates of the peptides from the F-actin surface. This approach is called transfer NOE and has been widely used to determine the structure of small ligands in complex with high molecular weight binding partners[45]. In addition, we recently showed that the transfer NOE effect can be used to analyze structural changes that occur upon binding of Tau peptides to microtubules[46].

Two-dimensional NOESY experiments with five different mixing times ranging from 50 up to 250 ms were recorded for Tau(254–290) in the presence of F-actin. Analysis of NOE signal intensities as a function of mixing time suggested that spin diffusion did not significantly contribute to the observed NOE intensities (Supplementary Fig. 6a). Figure 3b shows part of the amide-amide region of a NOESY spectrum of Tau(254–290) in the absence (red) and presence (blue) of F-actin. Because of the binding of the peptide to F-actin, additional NOE cross-peaks were observed (Fig. 3b; Supplementary Fig. 7). The NOE peaks, which were observed only in the presence of F-actin, provide experimental restraints for the structure of Tau(254–290) in complex with F-actin.

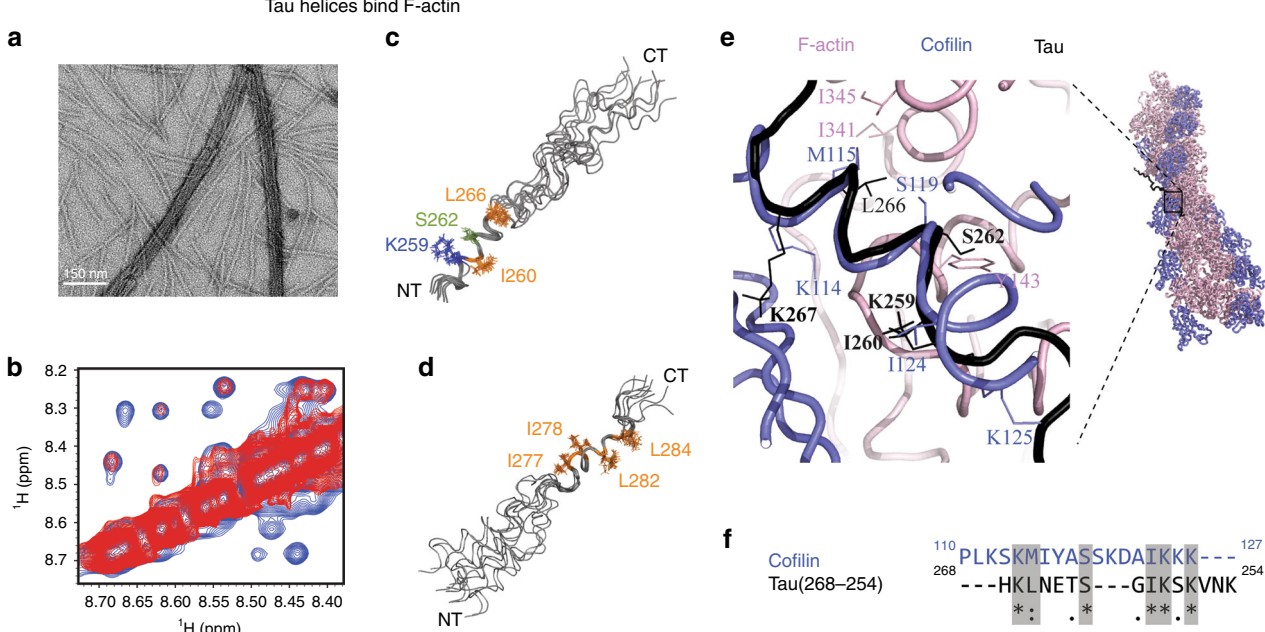

**Fig. 3** Tau(254–290) adopts helical structure upon binding to filamentous actin. **a** Electron micrograph demonstrating bundling of F-actin in presence of Tau(254–290). **b** Amide–amide region from 2D NOE spectra of Tau(254–290) in the presence (blue) and absence (red) of F-actin. **c** Ten lowest-energy conformers of F-actin bound Tau(254-290). Structures were aligned from N255 to H268. Hydrophobic residues are colored orange. In addition, K259 (blue) and S262 (light green) are highlighted. **d** Ten lowest-energy conformers of F-actin bound Tau(254–290) aligned from residue Q276 to V287. **e** Crystal structure of cofilin (blue) in complex with F-actin (pink; PDB id: 3JOS), superimposed onto the F-actin bound form of Tau(254–290) as derived from protein–protein docking (black). Side chains of selected residues are shown. **f** Sequence alignment of cofilin with residues 268–254 of Tau

The NOE cross-peaks were sequence-specifically assigned using 2D total correlation spectroscopy (TOCSY) spectra (Supplementary Table 1). On the basis of the resonance assignment, several medium-range, but no long-range contacts were identified (Supplementary Fig. 7). To obtain insight into the conformation that is responsible for the observed medium-range contacts, we analyzed cross-peaks that are characteristic for regular secondary structure (Supplementary Figs. 6b and 7). The analysis showed that the N- and C terminus of Tau(254–290) remain unstructured in complex with F-actin. In contrast, medium-range NOE cross-peaks, which are specific for secondary structure, were observed for residues ~260 to ~268 and residues ~277 to ~283. The corresponding residues in full-length Tau experienced PRE broadening (Fig. 2e; Supplementary Figure 4c).

Subsequently, experimental NOE contacts were used to calculate structural models of Tau(254–290) in complex with F-actin. To minimize NOE contributions from unbound molecules, intra-residual and sequential NOE contacts were excluded from the calculations. Thus, a total of 182 restraints went into the structure calculation, with 1.8 and 6.0 Å as lower and upper distance limits, respectively (Supplementary Fig. 6d). The use of such broad distance limits is required, because an accurate calibration of NOE intensities is not possible. The 10 lowest-energy conformations are presented in Fig. 3c, d. Analysis of secondary structure in the lowest energy conformation using STRIDE[47], identifies α-helix for residues 261–268 and 3–10 helix for residues 277–280. In some of the other conformers of Tau (254–290), however, residues 261–265 were identified not as α-helix but as 3–10 helix, while residues 277–283 were assigned to α-helix. The analysis suggests that the experimental NOE data witness the formation of helical structure in these regions upon binding to F-actin, but the number of detected restraints is not sufficient to define a unique conformation. In addition, the relative orientation of the two helical regions (residues 261–268

and 277–280) is not defined due to the lack of long-range contacts. Instead, the presence of a proline residue followed by three glycines in between the two helical regions suggests that the intervening residues remain flexible.

**Structural model for the Tau/F-actin binding site.** To derive a structural model for the Tau/F-actin binding site, we integrated the information that was obtained in the current work (Figs. 1–3) with the 3D structure of cofilin-decorated F-actin (Fig. 2a). To this end, the lowest energy conformation of Tau(254–290) was docked to the solvent-exposed, hydrophobic pocket of actin within the actin filament structure using the software Haddock[48]. Haddock clustered 192 complex structures in four clusters, representing 96.0% of the water-refined Haddock models. Within the highest scoring cluster (Supplementary Fig. 8), the N-terminal helical region of Tau(254–290) was bound to the hydrophobic pocket of F-actin (Fig. 3e).

The docking model of the Tau(254–290)/F-actin complex was compared with the 3D structure of cofilin-decorated F-actin (Fig. 3e). The comparison showed that the helical region of Tau (254–290) is positioned in the binding pocket in a similar way as the recognition helix of cofilin (blue). In this model, L266 of Tau (254–290) is 4 Å away from I345 of F-actin, while the HE atom of the cofilin residue M115 is 3.1 Å separated from the HG2 atom of actin's I341. In addition, I260 of Tau(254–290) was found in a similar location as I124 from cofilin. In agreement with the structural similarity, the involved sequences from cofilin and Tau can be aligned (Fig. 3f). Optimal sequence alignment was achieved, when the sequence of Tau was inverted (Fig. 3f), suggesting an inverted orientation of its recognition helix when compared to cofilin (Fig. 3e; Supplementary Fig. 9). Although a three-residue gap is present in the alignment (Fig. 3f), this gap appears to be compensated by a more extended structure of Tau

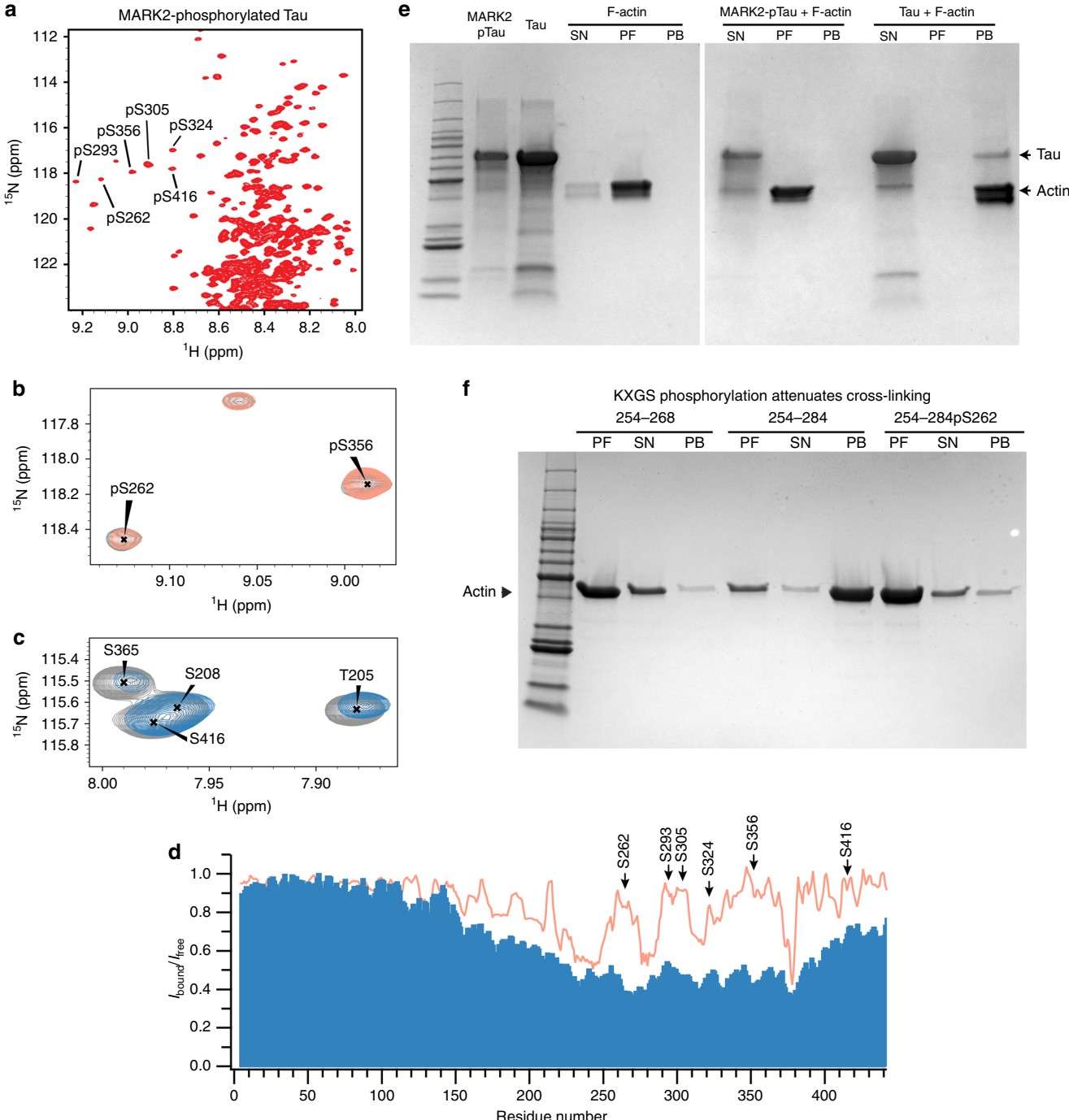

**Fig. 4** MARK2 phosphorylation decreases Tau's affinity for F-actin. **a** $^1$H-$^{15}$N HSQC of MARK2-phosphorylated Tau. Phosphorylated serine residues are labeled. **b** Selected region from a $^1$H-$^{15}$N HSQC of MARK2-phosphorylated Tau in the absence (gray) and presence of F-actin (orange; Tau/F-actin molar ratio of 1:1.5). **c** Selected region from a $^1$H-$^{15}$N HSQC of non-phosphorylated Tau in the absence (gray) and presence of F-actin (blue; Tau/F-actin molar ratio of 1:1.5). **d** NMR signal broadening induced by F-actin in non-phosphorylated (blue bars) and MARK2-phosphorylated Tau (orange line). Tau residues phosphorylated by MARK2 are highlighted. **e** Effect of MARK2-phosphorylation on the ability of Tau to bundle actin filaments. **f** Phosphorylation of S262 decreases the ability of Tau(254–284) to bundle F-actin

(black in Fig. 3e) when compared to the α-helix conformation of residues S119-A123 of F-actin bound cofilin (blue in Fig. 3e).

The docking model (Fig. 3e) suggests an important role of the Tau residue L266 for binding to F-actin. We therefore prepared a mutant version of Tau(254–290), in which L266 was replaced by a

glutamate, i.e., Tau(254–290)-L266E. We then used saturation transfer difference (STD)[49] experiments to probe the influence of the mutation on binding of Tau(254–290) to F-actin. When compared to the wild-type peptide, Tau(254–290)-L266E showed a decreased STD signal (Supplementary Figure 10c–e), indicating

that the mutation attenuated F-actin binding. Moreover, no additional cross-peaks were observed in the NOESY spectrum of Tau(254–290)-L266E in the presence of F-actin (Supplementary Fig. 10a, b), i.e., no transfer NOE effect occurred. In agreement with a decrease in affinity of Tau(254–290)-L266E for binding to F-actin (Supplementary Fig. 1), the mutant peptide was less efficient in promoting F-actin bundling (Supplementary Fig. 10f).

**Two helical regions in Tau are required for actin bundling.** Two helical regions, which are separated from each other by flexible residues, are formed in Tau(254–290) upon binding to F-actin (Fig. 3c, d). The two helices might be formed simultaneously, when a single molecule binds to F-actin, potentially folding in a cooperative manner. Alternatively, individual Tau (254–290) molecules can bind to F-actin either through residues ~260 to ~268 or through residues ~277 to ~283, while the other region will not be bound. Because of the ensemble nature of NMR measurements, the structures shown in Fig. 3 do not distinguish between these two mechanisms. In addition, other segments of Tau (Fig. 2b, e) are likely to change conformation upon binding to F-actin. To gain insight into these different mechanisms, we prepared the peptides Tau(292–319) and Tau(254–268).

Tau(292–319) comprises the residues of repeat R2 and R3, which experience strong signal attenuation upon addition of F-actin to full-length Tau (Fig. 1e, f). In agreement with the ability of Tau(292–319) to bind to F-actin, the peptide promotes bundling of actin filaments (Supplementary Fig. 5c). NOE experiments of Tau(292–319) in the absence and presence of F-actin revealed NOEs specific for the Tau(292–319)/F-actin complex (Supplementary Fig. 11). Sequential assignment of Tau (292–319) (Supplementary Table 2) showed that the F-actin specific NOEs are medium-range (Supplementary Figs. 11 and 12), which were subsequently submitted to structure calculation. Supplementary Figures 12a–c illustrate the resulting lowest energy conformations. Analysis of secondary structure using STRIDE[47] indicated that there is quite some variation in the location and identity of helical regions across the 20 lowest energy structures (Supplementary Figure 12a, b). In most structures a 3–10 helix was identified for residues 315–318, although in the lowest energy structure residues 315–318 are in α-helical conformation (Supplementary Figure 12c). The transfer-NOE data thus point to the formation of helical structure in Tau (292–319) upon binding to F-actin, but the number of detected restraints was not sufficient to define a unique conformation.

To address the importance of having multiple helical regions separated by flexible linkers for binding to and bundling of F-actin, we used the peptide Tau(254–268). Tau(254–268) contains the residues, which form the helix in the N-terminal half of Tau (254–290) (Fig. 3c), but not the C-terminal helical region (Fig. 3d). Sequential resonance assignment of the peptide (Supplementary Table 3), followed by NOE analysis (Supplementary Fig. 13) and structure calculations (Supplementary Fig. 14) showed that the short peptide folds into an α-helix (residues 259–265) in complex with F-actin that is similar to the one found in Tau(254–290) (Supplementary Fig. 14b). In contrast to Tau(254–290), however, Tau(254–268) was not able to bundle F-actin (Supplementary Fig. 5d). Thus, two Tau interaction sites have to be present, in order to bundle F-actin filaments.

**Phosphorylation attenuates Tau/F-actin interaction.** Tau protein is post-translationally modified by phosphorylation, acetylation and several other modifications[26, 50–52]. An important class of kinases that phosphorylate Tau at S262 and the other

KXGS motifs in the repeat domain are the microtubule-associated protein/microtubule affinity-regulating kinases (MARKs)[53–55]. We therefore phosphorylated full-length Tau by MARK2. The downfield chemical shift of phosphorylated residues (Fig. 4a) is in agreement with previous reports and confirms phosphorylation at S262, S293, S305, S324, S356, and S416[54]. Subsequently, $^1$H-$^{15}$N HSQC spectra of MARK2-phosphorylated Tau were performed in the absence and presence of F-actin (Figs. 4b, c). In contrast to the non-phosphorylated protein, little attenuation of NMR signals in vicinity of the phosphorylation sites was detected (Fig. 4b–d). Residues such as S262 and S356 largely retained the NMR signal intensity in presence of F-actin (Fig. 4b). However, residues in the proline-rich region, as well as V275-L284, which do not contain MARK2-phosphorylation sites, experienced similar F-actin induced signal attenuation when compared to non-phosphorylated Tau (Fig. 4d).

The NMR experiments demonstrate that MARK2-phosphorylation of Tau attenuates its binding to F-actin. Consistent with a reduced affinity, MARK2-phosphorylated Tau failed in bundling actin filaments (Fig. 4e). In addition, attachment of a phosphate group to S262 in the peptide Tau (254–284) decreased the affinity of the peptide for F-actin (Supplementary Fig. 1) and lowered the amount of Tau (254–284)-promoted actin bundles (Fig. 4f). In the NMR-based docking model with F-actin, S262 might share a hydrogen bond with Y143 of actin (Fig. 3e). Thus, phosphorylation within one of the seven interaction motifs of Tau (e.g. S262 in the helical segment formed by residues ~260 to ~268; Figs. 2 and 3c) weakens the interaction of this motif with F-actin, in agreement with the structural studies (Fig. 3).

## Discussion

Tau and other microtubule-associated proteins, such as MAP2, not only bind to microtubules but also to filamentous actin, which results in cross-linking and bundling of actin filaments (Fig. 1a)[3–10]. The interaction of Tau with actin is important for neurite outgrowth and synaptic dysfunction[11, 12, 22]. Pathogenic forms of Tau have been connected to the formation of actin-rich rods[56], which were found to induce neurodegeneration in *Drosophila* neurons[22]. In addition, Tau-induced neurotoxicity is associated with increased F-actin levels[22], and Tau-induced remodeling of the actin cytoskeleton can cause plasma membrane blebbing[57]. In agreement with previous findings, we show that Tau binds with high-affinity to filamentous actin, resulting in F-actin bundling (Fig. 1b). Both the proline-rich region and the microtubule-binding repeats contribute to the interaction (Figs. 1e, f). Interaction with the proline-rich region is primarily of electrostatic nature[58]. In contrast, short hydrophobic residue stretches in the repeat domain (Supplementary Fig. 3a) bind to the hydrophobic pocket between subdomain 1 and 3 of actin (Fig. 2). This hydrophobic pocket is solvent-accessible on the surface of actin filaments[59–61]. The F-actin-interacting Tau residues are separated from each other by flexible linkers, which enable a high degree of dynamics and multivalency in the Tau/F-actin interaction.

We further showed that the F-actin-interacting residues in the repeat domain of Tau, fold into short helices upon binding to filamentous actin (Fig. 3 and Supplementary Figs. 12 and 14). Folding of local regions of Tau into helical structure is consistent with the formation of α-helices in actin-binding proteins, in which the actin-interacting region is intrinsically disordered prior to binding to actin[59]. The Tau regions that fold into helical structure include part of $^{275}$VQIINK$^{280}$ (Fig. 3d), a hexapeptide

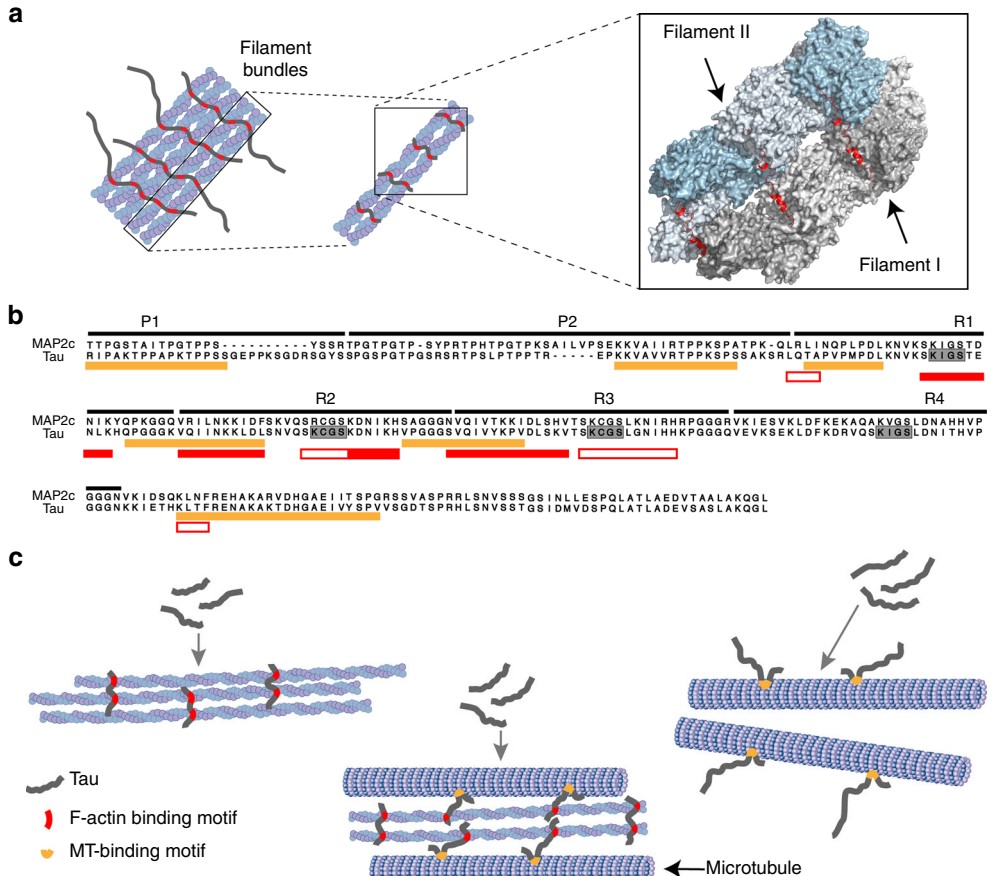

**Fig. 5** Model of Tau-driven actin bundling F-actin/microtubule network formation. **a** Schematic representation of cross-linking of actin filaments by helices of Tau, which are separated by flexible linkers. Each helix (shown in red) is bound to a hydrophobic actin pocket in a neighboring filament (gray and light blue). **b** Sequence alignment of the repeat domains of MAP2c and Tau. Tau residues, which bind to F-actin, are marked by red open bars, experimentally proven F-actin interacting regions by red filled bars, and microtubule-binding sites in orange. KXGS motifs are highlighted. **c** Because Tau contains multiple interaction sites for both F-actin (this work) and microtubules[72], Tau can cross-link the cellular cytoskeleton[7]. While short segments in the repeat domain fold into helical structure (red) in complex with F-actin, Tau residues in complex with microtubules form a microtubule-specific hairpin structure (orange[46])

that is important for aberrant aggregation of Tau into paired helical filaments[62]. The hexapeptide populates extended structure in solution[27], is found in cross-β structure in amyloid fibrils[63], but folds into helical structure in complex with F-actin (Fig. 3d). In addition, the N-terminal halves of each of Tau's four 18-residue aminoacid repeats[64] bind to F-actin's hydrophobic pocket and fold into helical structure (Fig. 3c; Supplementary Figs. 12 and 14). Consistent with the compatibility of these regions for helical conformation, Tau residues 253–260, 315–322, and 345–354 populate helical structure in the presence of detergent and upon binding to membranes[65]. Docking of experimentally determined Tau helices to the F-actin surface together with sequence alignment showed that the overall position of the interacting Tau and cofilin helices is well conserved, but the directionality of their polypeptide chain is inverted (Fig. 3e, f). The directionality of the Tau(259–267)-helix is similar to that of the actin-interacting helix of toxofilin (Supplementary Fig. 9), an actin-binding protein that is secreted into host cells during invasion[60]. Taken together the data show that Tau utilizes a structural mechanism for binding to F-actin that is used by actin-binding proteins.

Binding of Tau and MAP2 to filamentous actin results in actin cross-linking and filament bundling[5, 6, 14, 22, 40–43]. At the same time, binding to F-actin can be retained, but the cross-linking

activity disrupted. For example, a C-terminal fragment and a peptide corresponding to the second microtubule-binding repeat bind actin, but are incapable of mediating actin cross-linking[5, 17]. These findings suggest that multiple domains of Tau might be required for actin cross-linking. Consistent with this hypothesis, we showed that a single Tau helix is sufficient for binding to F-actin, but at least two helices, which are separated by a flexible linker, are required for cross-linking activity (Supplementary Figs. 5, 13 and 14). On this basis, we developed a model, in which two helical regions in Tau are bound to two different actin filaments (Fig. 5a). Because Tau contains seven actin-binding segments (Fig. 2e), up to seven actin filaments could be linked together by a single Tau molecule. Steric constraints are minimized by the presence of flexible linkers between the seven actin-binding segments. In addition, the excess positive charge, which is present in the repeat domain of Tau, will decrease electrostatic repulsion between the negatively charged filament surfaces. Because of the high sequence conservation in the repeat domain between Tau and other microtubule-associated proteins such as MAP2c (Fig. 5b), our data suggest that the described mechanism for actin cross-linking is more widely applicable.

The interaction between actin filaments and microtubules is important for cellular processes including cell division, vesicle

and organelle transport, axonal growth and migration[66]. Tau mediates interactions between microtubules and actin in neuronal growth cones, where Tau-mediated cytoskeletal interactions promote morphological changes. Moreover the tight connection between microtubules and actin drives the guided extension of axons during neuronal development as well as the formation and activity of synapses in mature neurons[67]. However the molecular mechanisms regulating this cytoskeletal crosstalk are poorly understood. Tau colocalizes with actin in differentiating PC12 cells[68], N2a cells[69] and in the postsynaptic compartments of mature neurons[70]. Tau has also been found on dynamic microtubules in the actin rich growth cone areas of developing neurons[71]. These data support a role for Tau as regulator of microtubules and actin and suggests its potential role as linker between the both cytoskeletons. Consistent with these findings in cells, Tau binds in vitro simultaneously to microtubules and actin filaments[7]. We previously showed that the Tau residues 168–185, 224–237, 245–253, 269–284, 300–313, and 375–398 bind to microtubules at the interface between tubulin heterodimers (Fig. 5b)[72]. Thus, when both F-actin and microtubules are present, they would compete for binding to residues 245–246, 275–284, and 305–313, while Tau residues 259–267, 289–297, and 320–330 will preferentially interact with F-actin (Fig. 5b). Notably, the conformations that are induced in Tau upon binding to F-actin and microtubules are very different, because F-actin interaction is connected to helix formation (Fig. 3), while short Tau regions fold into a hairpin-like structure in complex with microtubules[46]. Thus, the availability of multiple Tau sites for binding to microtubules and F-actin together with Tau residues, which are dedicated for binding to only F-actin or microtubules (Fig. 5b), suggests a model in which microtubules and actin filaments are cross-linked by Tau in a multivalent, dynamic manner (Fig. 5c), in agreement with experimentally observed cross-linking of microtubules and actin filaments by Tau[7].

Tau undergoes several post-translational modifications that influence its function, aggregation and toxicity[73]. An important post-translational modification is phosphorylation, because Tau is hyperphosphorylated in the brains of patients with Alzheimer's disease[74]. Phosphorylation of serine residues at KXGS motifs in the repeat domain of Tau, such as S262, strongly reduce the affinity of Tau for microtubules[75]. In addition, phosphorylation by endogenous protein kinases inhibits the ability of Tau and MAP2 to cross-link and bundle actin filaments[5, 6]. Consistent with these data, we showed that phosphorylation of Tau by MARK2 reduces the affinity of Tau for F-actin (Fig. 4d) and prevents F-actin bundling (Fig. 4e). In addition, specific phosphorylation at S262 decreased the F-actin affinity of Tau (254–284) (Supplementary Fig. 1) and decreased the peptides bundling activity (Fig. 4f). The KXGS motif, which contains S262, is located in one of the F-actin interacting helices of Tau (Fig. 3c). The side chain of S262 points towards actin's hydrophobic binding pocket (Fig. 3e), providing a structural explanation for the lower affinity of S262-phosphorylated Tau(254–284). While our as well as previous biophysical studies demonstrated that phosphorylation at the KXGS motifs decreases the affinity of Tau for both microtubules and F-actin, cellular studies showed that phosphorylation of the KXGS motifs cause MAP2 and Tau to localize to the actin cytoskeleton[9]. In addition, phosphorylated Tau is recruited to actin-cofilin rods, which might lead to neuropil threads[56]. Thus, phosphorylation of Tau at KXGS motifs fosters the interaction with actin-containing structures, but attenuates the direct interaction of Tau with microtubules and F-actin. Both findings together suggest that localization of phosphorylated Tau to actin-containing structures might be mediated

by additional actin-binding proteins. Future studies should be targeted toward identification of these proteins. In addition to phosphorylation, acetylation of Tau at K281, a residues that is located in an F-actin interacting helix of Tau (Fig. 3d), affects the interaction of Tau with actin and modulate Tau-mediated toxicity[26].

In summary, we provided insight into the interaction between actin filaments and microtubule-associated proteins and the crosstalk between the cytoskeletal networks formed by actin filaments and microtubules. We identified Tau residues that are important for complex formation and provided high-resolution information about the structural changes that occur in Tau upon binding to filamentous actin. We showed that Tau utilizes a structural mechanism for binding to actin filaments, which is used by many actin-binding proteins, but allows a highly dynamic and multivalent interaction with both actin filaments and microtubules. The multivalent interaction of Tau with F-actin and microtubules results in cross-linking of the cellular cytoskeleton. Our results also help in dissecting an important pathogenic mechanism in AD.

## Methods

**Protein preparation.** Human Tau (2N4R) was expressed in the *E. coli* strain BL21 (DE3) (MERCK Millipore) using the vector pNG2 as described[13]. The bacterial pellet was resuspended in lysis buffer (50 mM Mes, 500 mM NaCl, 1 mM MgCl$_2$, 1 mM EGTA, 5 mM DTT, pH 6.8) and supplemented with a protease inhibitor mixture. Cells were disrupted using a French pressure cell and subsequently boiled for 20 min. The soluble extract was isolated by centrifugation and the supernatant was dialyzed twice against FPLC buffer (20 mM Mes, 50 mM NaCl, 1 mM EGTA, 1 mM MgCl$_2$, 2 mM DTT, 0.1 mM PMSF, pH 6.8), and loaded onto a FPLC SP-Sepharose column. Proteins were eluted using a linear gradient of elution buffer (20 mM Mes, 1 M NaCl, 1 mM EGTA, 1 mM MgCl$_2$, 2 mM DTT, 0.1 mM PMSF, pH 6.8). Tau breakdown products were separated in a second chromatography step by using a Superdex G200 column (GE Healthcare) with the separation buffer (137 mM NaCl, 3 mM KCl, 10 mM Na$_2$HPO$_4$, 2 mM KH$_2$PO$_4$, pH 7.4, 1 mM DTT). Protein samples uniformly enriched in $^{15}$N were prepared by growing *E. coli* bacteria in minimal medium containing 1 g l$^{-1}$ of $^{15}$NH$_4$Cl.

Synthetic peptides were purchased from EZBiolab USA or synthesized in house on ABI 433A (Applied Biosystems) and Liberty 1 (CEM) machines. Peptides were synthesized with acetyl and amide protection groups at the N- and C terminus, respectively. Peptides were purified by reversed-phase HPLC, and the pure product was lyophilized.

Non-muscle human actin (catalog no. APHL99) as well as human cofilin (catalog no. CF01) were bought from Cytoskeleton, Inc. Hundred micrograms of cofilin powder were resuspended in 20 µl of distilled water to have a final buffer of 10 mM Tris pH 8.0, 10 mM NaCl, 5% sucrose and 1% dextran. The sample was then dialyzed against General Actin Buffer (GAB; 5 mM TrisHCl pH 8.0, 0.2 mM CaCl$_2$, 0.2 mM ATP, 0.5 mM DTT) or NMR buffer (50 mM NaH$_2$PO$_4$/Na$_2$HPO$_4$ pH 6.8, 10 mM NaCl, 1 mM DTT, 10% (v/v) D$_2$O) to be used for electron microscopy or NMR experiments, respectively. In case of actin, lyophilized actin was reconstituted to 10 mg ml$^{-1}$ by adding 100 µl of distilled water to be in the following buffer: GAB with 5% (w/v) sucrose and 1% (w/v) dextran. Subsequently, the sample was spun at 18,400 × *g* for 15 min, the supernatant was kept, and actin was polymerized by incubating 0.4 mg ml$^{-1}$ G-actin in GAB and supplemented with polymerization buffer (1/10th the volume; 100 mM Tris HCl pH 7.5, 20 mM MgCl$_2$, 500 mM KCl, 10 mM ATP) for 1 h at room temperature. Afterwards, the sample was centrifuged at 100,000 × *g* during 1 h at 4 °C. Filaments were collected at the bottom of the ultracentrifuge tube. The pellet was then resuspended in GAB or NMR Buffer as required.

For labeling actin protein with MTSSL ((1-oxy-2,2,5,5-tetramethyl-d-pyrroline-3-methyl)-methanethiosulfonate, Toronto Research Chemicals), DTT was removed from the sample by using a 2 ml Zeba desalting column previously equilibrated with GAB at pH 6.8. Free sulfhydryl groups were then modified by incubating the sample overnight at 0 °C with a 20-fold molar excess of MTSSL. Unreacted MTSSL was removed with another 2 ml Zeba desalting column. Subsequently, MTSSL-labeled actin was polymerized as described above, in order to have MTSSL-labeled F-actin with the MTSSL molecule covalently bound to Cys374.

**F-actin binding assay.** NBD-conjugated actin from α-skeletal muscle was bought from Hypermol (Bielefeld, Germany). In this experiment, 7-chloro-4-nitrobenz-2-oxa-1,3-diazole (NBD)-labeled G-actin was polymerized starting from 0.4 mg ml$^{-1}$ of the conjugated actin supplemented with 10-fold excess of phalloidin (catalog no. P2141, Sigma-Aldrich) to provide stable filaments. Subsequently, 0.25 µM of

labeled F-actin was incubated with increasing amounts of Tau or Tau fragments. The change of fluorescence intensity from NBD was followed in a Cary Eclipse fluorescence spectrophotometer (Agilent Technologies, Inc.). Excitation and emission were set at 480 and 530 nm, respectively, and experiments were carried out at 25 °C in GAB. As reported by Hertzog, M. et al.[31] the normalized decrease in fluorescence (equation 1) was analyzed and fitted by using equation 2.

$$E = \frac{(F - F_0)}{(F_{max} - F_0)} \qquad (1)$$

$$E = \frac{1}{2}C + \frac{1}{2}Z - \frac{1}{2}\sqrt{(C + Z)^2 - 4Z} \qquad (2)$$

with

$$Z = \frac{[partner]}{[F-actin]} \text{ and } C = 1 + \frac{K_d}{[F-actin]}$$

**Electron microscopy**. Samples were bound to a glow discharged carbon foil covered copper grid. After washing of the grids using ddH$_2$O, the samples were stained using 1% uranyl acetate. The samples were evaluated at room temperature using a CM 120 transmission electron microscope (FEI, Eindhoven, and The Netherlands) and a TemCam F416 CMOS camera (TVIPS, Gauting, Germany).

**Protein co-sedimentation**. Mixtures of F-actin with Tau (1:1) or Tau peptides (1:10 molar ratio) were incubated at room temperature for 30 min. In case of the shorter peptide Tau(254–268), a 1:30 ratio was used. Centrifugation was performed in two steps. First, a spin rate of $5000 \times g$ was used during 15 min to collect bundles of F-actin (PB) at the bottom of the tube. Secondly, the remaining sample was centrifuged at $100,000 \times g$ during 1 h collecting actin filaments (PF). The final supernatant (SN) was also kept for later usage. Pellets were resuspended in 80 µl of GAB. A volume of 25 µl of sample were taken for EM while the remaining 55 µl were loaded in a 4–20% gradient SDS gel, which was colored with Bromophenol Blue and denatured with SDS and heat.

**NMR spectroscopy**. NMR spectra were acquired on Bruker Avance800 and 900 MHz NMR spectrometers at 278 K to reduce solvent exchange. 2D $^1$H-$^{15}$N HSQC spectra consisted of 64 scans, spectral windows of 12 ppm in the proton and 25 ppm in the nitrogen dimension. 2D NOESY experiments were acquired using 88 scans, 12 ppm spectral width and a time domain of 512 and 2048 points in the first and second dimension, respectively. To generate buildup curves, mixing times of 50, 80, 100, 150, and 250 ms were used.

For HSQC-based competition experiments, 20 µM of $^{15}$N-labeled Tau were mixed with 30 µM F-actin in NMR buffer supplemented with 0.2 mM ATP and 0.2 mM CaCl$_2$ (sample 2). Similarly 20 µM of $^{15}$N-labeled Tau was prepared in absence of F-actin as reference (sample 1). Finally, a 10-fold excess of cofilin, previously dialyzed in NMR buffer overnight, was added to sample 2. Competition was monitored by reading out the variation in signal broadening in 2D $^1$H-$^{15}$N HSQC spectra of Tau.

For NOESY experiments, 27 µM of F-actin were mixed with 800 µM of Tau peptide in NMR buffer supplemented with 0.2 mM ATP and 0.2 mM CaCl$_2$. A reference experiment was performed in similar conditions only with 800 µM peptide. For resonance assignment, a 2D TOCSY experiment with a mixing time of 75 ms was performed. All data were processed using NMRPipe[76] and analyzed using Sparky[77].

**Structure calculation**. Distance restraints were obtained from cross-peaks observed in 2D $^1$H-$^1$H NOESY spectra acquired with 100 ms mixing time. A total of 200 conformers were calculated using the standard simulated annealing schedule with 10,000 torsion angle dynamics steps per conformer using CYANA 3.97[78]. Subsequently, structures and distance restraints derived from CYANA were transformed to the XPLOR-NIH input file format. Final structures were refined in XPLOR-NIH using a restrained simulated annealing (SA) protocol[81]. During simulated annealing, the temperature was lowered from 500 K down to 100 K. To minimize the influence of calibration of NOE intensities, only medium-range NOE contacts were used with lower and upper limits set to 1.8 and 6.0 Å, respectively. Statistical knowledge-based potentials for 2D and 3D torsion angle correlations were implemented into the SA protocol and their non-dimensional force constant was ramped during the refinement from 0.002 to 10. A total of 400 structures were calculated and the 20 lowest-energy conformers were selected for analysis using the protein structure validation software suite (PSVS)[79]. Visualization was performed using PYMOL (The PyMOL Molecular Graphics System, Version 1.5.0.4 Schrö-dinger, LLC) and MOLMOL[80]. The same protocol for structure calculation and analysis was followed for all three peptides.

**Data availability**. Structures of Tau(254–268), Tau(254–290), and Tau(292–319), when bound to F-actin, were deposited in the ProteinDataBank (PDB id: 5NVB, 5N5A, and 5N5B). Additional relevant data are available from the corresponding author upon reasonable request.

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

## Acknowledgements

We thank Karin Giller for MTSSL-labeling of actin, Kerstin Overkamp for peptide synthesis, and Sabrina Hübschmann for preparation of Tau samples. Y.C.F. thanks Dr. Javier Oroz and Dr. Nasrollah Rezaei-Ghaleh for useful discussions. This work was

supported by the Deutsche Forschungsgemeinschaft through project ZW 71/8-1 (to M.Z.).

## Author contributions

Y.C.F. performed and analyzed NMR and biophysical experiments and performed structure calculations. H.K. helped with analysis of NMR data and structure calculations. J.B. prepared Tau samples. D.R. recorded electron micrographs. E.M. supervised Tau preparation. Y.C.F., J.B., E.M., and M.Z. wrote the paper. M.Z. designed the project.

## Additional information

**Competing interests:** The authors declare no competing financial interests.

