## [Peer Review File · Nature Communications]

Reviewers' comments:

Reviewer #1 (Remarks to the Author):

This manuscript describes studies of the interactions between tau and filamentous actin. EM and co-sedimentation are used to show that actin filaments are bundled in the presence, but not absence, of tau and contain tau. Tau affinity for f-actin is measured using NBD-labeled f-actin fluorescence. Binding sites for f-actin on tau were localized using solution state NMR signal intensity loss, and tau binding sites on f-actin were localized using competition with cofilin binding, monitored by NMR, as well as by PRE resulting from a spin labeled f-actin. Tau fragment K18 was also shown to bind to and bundle f-actin. Smaller tau fragments corresponding to regions broadened upon f-actin binding were used to analyze the structure of small regions of tau bound to f-actin. Transferred NOEs obtained for a fragment corresponding to the end of R1 and the beginning of R2 (254-290) indicated helix formation in regions 259-267 and 275-284 and structure calculations accordingly resulted in helical structure in these regions. A second fragment corresponding to the end of R2 and beginning of R3 (292-319) gave similar results with helical structure located at its termini. A regions corresponding to only one of helices of the 254-290 fragment (254-268) also bound in a helical conformation, but unlike the two-helix fragments, was inactive in a bundling assay. The structural model for the 254-290 fragment was then docked into the cofilin-binding site of f-actin using HADDOCK. The result shows binding that is similar in nature to that of cofilin, but reversed in direction. Finally, phosphorylation by MARK2 was shown to decrease actin binding at the relevant phosphorylation sites, but not at distant sites, and fragment 254-290 phosphorylated at site 262 was shown to reduce bundling.

This is an interesting paper that provides the first detailed structural information regarding the interactions of tau with f-actin, and hence is of significance. That said, the paper relies entirely on in vitro observations, which is reasonable for obtaining the structural data, but which handicaps the paper in terms of functional significance and a more general impact. Indeed, despite the number of references to previous work on tau-actin interactions, the physiological significance of these interactions is perhaps not widely appreciated. Data supporting a physiological significance of the observed structural features/interactions would be most appropriate. Beyond this limitation, there are a number of technical issues (below) that indicate an inattention to detail, which further reduces the impact of the paper as is. Many of these should be fixable without undue effort, but the issue of functional/physiological assays and relevance may be more difficult to address.

1. Binding curves for all data in Fig. S2 should be included.
2. A zoomed in version of the PRE data should be provided that includes only the regions with attenuated signals so the location of the binding sites can be visualized more easily to confirm/assess the described boundaries (L243-A246, K259-K267, V275-L284, S289-S293, S305-V313, S320-H330 and K375-F378).
3. The methods section appears to only include a description of the structural calculation for a single (presumably the 254-290) peptide. Details for the structure calculations for all three peptides should be explicitly included (if they are all the same, this should be indicated).
4. The resonance assignments obtained from the 2D Tocsy experiments for all three peptides should be included as a table.
5. The NOE patterns for the 292-319 fragment are quite different from those typically observed for helical structure, which is defined by $i-i+3$ and $i-i+4$ restraints. In addition, the quality of the calculated structures is quite low, with significant populations in the generously allowed and disallowed regions of the Ramachandran plot. In this case, one wonders to what extent the calculated structure actually represents the actual structure. I think a more detailed frank description of the structure calculation results and their quality is required.
6. The data used for the calculation of the isolated helix fragment (254-268) should be included, along with information regarding the structure calculation/validation, as for the other two fragments (i.e. as for figures S4 and S5).

7. For all calculated structures, an indication should be provided of the criteria used to identify alpha-helical structure, and the boundaries of such structure. The structural ensembles exhibit structure that looks rather atypical for alpha-helix, and this should be clarified. The authors should keep in mind that many non-experts may read their work, and providing a vague description of the structural results will result in a misinterpretation of the results.
8. For all calculated structures, PDB files should be deposited in the protein data bank.
9. The true test of the validity and relevance of structural complexes is structure-function studies and these are curiously missing from this work. The conclusions would be substantially strengthened if the structural work was used to design mutations that could perturb the tau-actin interaction and results in quantifiably altered function (at a minimum in bundling assays, much better yet in some in situ or in vivo assay of the significance of tau-actin interactions).
10. In the discussion, the authors indicate that toxofilin, another f-actin binding protein, binds in a similar inverted orientation as the 254-290 fragment. It would be informative to include a comparison/overlay with the toxofilin complex as an additional supplementary figure.
11. It seems only natural that the bundling ability of MARK2-phosphorylated tau should be assayed and described. I'm surprised this wasn't included.
12. In the discussion, the authors write 'Interaction with the proline284 rich region is primarily of electrostatic nature'. They should remind the readers that this is based on Fig. S1, as this is only very briefly mentioned in the results, and not explicitly interpreted in this manner.
13. The authors propose a competition between microtubules and actin for certain tau binding sites. It would be nice to see this tested/demonstrated experimentally. The authors note that flexible regions connect the different binding sites, but the size of these filamentous assemblies is quite large, and it is unclear to what extent the sterics would allow the proposed cross-linking, especially between near-by sites.

Reviewer #2 (Remarks to the Author):

This paper by Fontela and colleagues presents a detailed structural model for the binding of tau to F-actin. The model is based largely on experiments in which heteronuclear single quantum coherence spectroscopy was used to analyze F-actin-binding tau fragments in the presence and absence of F-actin. Key features of the model include: 1) tau binds to a hydrophobic pocket between subdomains 1 and 3 of polymerized actin subunits; 2) a single short helix within the microtubule-binding repeat (MTBR) domains of tau is sufficient for F-actin binding; 3) at least 2 out of 7 such helices are required for bundling of actin filaments by tau; 4) tau phosphorylation at S262 reduces binding to F-actin; 5) some, but not all tau binding sites for F-actin overlap with tau binding sites for microtubules; and 6) tau should be able to cross-link actin filaments to microtubules.

On the whole, this is an impressive study that points to both similarities and differences in the mechanisms by which tau binds to F-actin versus microtubules. It should be noted, however, that the physiological significance of tau interactions with actin filaments in vivo, including the extent, if any, to which such interactions occur in normal and diseased states, is not well established. With that caveat in mind, the crucial involvement of tau in the pathogenesis of tauopathies, including Alzheimer's disease, indicates that this study is bound to attract the attention of scientists who are interested in macromolecular structures germane to neurodegeneration.

One issue that deserves further attention concerns Figure 5C, which shows how tau can hypothetically cross-link actin filaments to microtubules. This figure should be supported by experimental evidence, or the figure and all discussion of it should be removed. The only other suggestions I have concern the following minor issues.

1) It would be helpful to readers if all figures were so self-explanatory that shifting attention away from the figures in order to read the figure legends would be unnecessary. To that end, please directly label the positions of tau and actin on Figure 1c; of tau, actin and cofilin on Figure 2c; and of actin on Figure 4c. Also, please indicate the lengths of the scale bars directly on Figures 1b and 3a.

2) The NBD-actin fluorescence change assay is not widely known and is not referenced. Please add the original reference for this technique.

3) The following sentence on page 4 is somewhat misleading: "After addition of cofilin, Tau's NMR resonance intensities and positions were similar to those in the absence of F-actin (Fig. 2b), indicating that (i) Tau is no longer bound to F-actin and (ii) cofilin does not bind to Tau.". While the statement is consistent with the data shown, it neglects to consider the possibility that the cofilin severed the actin filaments, which then partly depolymerized, which in turn would lead to decreased tau binding to F-actin.

4) Figure S3 has 3 parts: (a), (b) and (c), but part (b) is labeled as (a).

5) The following statement from the bottom of page 8 to the top of page 9 seems incorrect: "in agreement with the sequence similarity of Tau(292-319) (corresponding to part of R1/R2) with Tau(254-290) (corresponding to part of R2/R3).....". The sentence should state that Tau(292-319) corresponds to R2/R3, while Tau(254-290) corresponds to R1/R2.

6) On page 9, the term "wild type" is awkwardly used to refer to wild type tau that had not been phosphorylated in vitro by MARK2. Please change "wild type" to "non-phosphorylated".

Reviewer #3 (Remarks to the Author):

The article by Fontela, et al., presents NMR and other data on the mechanism of interaction between the intrinsically disordered protein, Tau, and F-actin. This interaction is associated with Tau-dependent bundling of F-actin, which is thought to play a role in synaptic structure. In summary, the authors report that short motifs that are repeated several times in the central "repeat region" of Tau individually bind to a surface exposed pocket on actin subunits within F-actin; this situation enables multivalent binding of Tau to F-actin, which the authors argue promotes F-actin bundling. The authors mapped binding sites within Tau for F-actin using several NMR methods, including mapping peak intensity losses in 2D spectra for ¹⁵N-Tau due to binding to F-actin. In addition, they MTSL spin-labeled actin monomers in F-actin at C374 and monitored peak intensity changes in spectra of bound ¹⁵N-Tau to map where on F-actin Tau binds. These experiments, while informative, were limited by the disappearance of Tau resonances when it bound to the large F-actin polymer. To overcome this limitation, the authors studied several truncated forms of the F-actin binding region of Tau that were prepared by peptide synthesis. The peptides, with fewer binding sites, exhibited lower affinity for F-actin and enabled characterization of the actin-bound conformation using a method called "transfer NOE". Using this approach, the authors determined that the actin-binding motifs adopt a helical conformation with conserved hydrophobic residues proposed (based on modeling/docking) to engage a pocket on actin subunits that is also engaged by cofilin. Importantly, the authors showed that cofilin and the Tau motifs compete for the same site on actin. Overall, the manuscript is well written and the NMR and other data are of high quality. This reviewer accepts the authors' general conclusions but would like to see the sequence-specificity of Tau motif/actin interactions tested through mutagenesis. Also, it would be nice to have interaction mapping data from at least a second MTSL-labeled form of actin to complement the single dataset that utilizes Cys 374. These additional experiments are needed to bolster the authors' conclusions prior to further consideration for publication. There are also

numerous technical points to address, as follows.

1. Suppl Fig 1 does not address hydrophobicity, as noted on page 4, line 105. The authors should include an analysis of hydrophobicity for Tau in order to discuss this issue.
2. Switching between I/I0 and (1-I/I0) formats for presenting NMR binding site mapping data is confusing. It is suggested that a single format should be used throughout the manuscript.
3. Control lanes are lacking in Fig. 2c showing migration of the three individual components in the experiment. These control lanes need to be included in the gel.
4. The authors should perform a titration of F-actin into 15N-Tau to determine local Kd values for residues within the various binding motifs. Do these values reflect the value from FP? This relates to comment #6, below.
5. The authors state that only one of six Cys residues (C374) is labeled with MTSL. Is this known experimentally? Certainly Cys 374 is highly accessible to solvent. The authors should compute the solvent accessibility of the other five to support their argument that they are inaccessible. How was MTSL labeling confirmed? What was the stoichiometry of labeling? Mass spectrometry could be used to address this issue. Some type of analytical data need to be provided to document MTSL labeling at Cys 374.
6. Figs. 1e & f, 2b & e. The authors state that the resonances of 15N-Tau are broadened beyond detection upon binding to F-actin polymers; consequently, peaks do not shift upon binding but rather disappear. With a Kd value of 60 nM (from FP), the fraction of free Tau under the conditions of the NMR experiments would be smaller than 20% but 20% to 40% of the starting signal persists in the presence of F-actin according to these figures. What is the explanation for the residual NMR signals? The authors should perform a titration of F-actin into 15N-Tau to determine residue-specific Kd values and determine whether they agree with the FP-derived value. The different actin binding motifs of Tau probably behave independently and may exhibit weak local Kd values but may function collectively to give the observed overall tight binding—a type of dynamic complex. The authors should clarify the situation and provide a more quantitative description of this binding event based on NMR data.
7. Fig. 1e. The authors should show the full 2D HSQC spectrum as a supplemental figure.
8. Page 5, Line 148. “Notably, no PRE effect was observed at the C-terminus.” What residues constitute the “C-terminus”? There are PRE effects between residues 380-400. Please clarify and revise, if needed.
9. Pages 5-6. Please define the residue boundaries of the Tau K18 construct.
10. Page 6. What is the binding constant of the residue 254-290 Tau peptide for F-actin? This should be reported.
11. Fig. 3b. Please show zoom views of key regions of the 2D NOESY spectrum as Supplemental figures, including the full amide region and the H α -HN region. Also, please display these with symmetric F1 & F2 dimension limits/scales, as is the usual convention.
12. Page 7, top. “In contrast, NOE cross-peaks, which are specific for α -helical conformation, were observed for residues 259-267 and residues 275-284. The corresponding residues in full-length Tau experienced pronounced PRE broadening (Fig. 2e), in agreement with binding to F-actin’s hydrophobic pocket.” The PRE profile in Fig. 2e appears to extend beyond residue 284 as far as almost to residue 300. Why did the authors not investigate a slightly longer Tau peptide?
13. Fig. 3f. It is difficult to understand how the inverted Tau 254-268 region could bind to F-actin the same way as cofilin in the noted PDB file because the 110-127 region of cofilin forms a continuous helix while Tau has a three residue long gap in the sequence alignment. Can the authors please comment on this and revise the manuscript, if needed.
14. Suppl. Table S1. The authors should include all of the Haddock clusters and their respective scores/statistics. What restraints were used in the docking experiments? A restraints violation energy is given. How have the authors validated their structural model? They should perform mutagenesis of residues in Tau that appear to be critical for binding to F-actin and i) demonstrate loss of binding by NMR, and ii) altered docking results using Haddock.

15. Suppl. Fig. 5. The authors should illustrate regions of the 2D NOESY spectra for the Tau(292-319) peptide from which structural restraints were extracted (as discussed in comment # 11, above).
16. Page 9. "We phosphorylated full-length Tau by MARK2, followed by NMR measurements in the absence and presence of F-actin. In contrast to the wild-type protein, little perturbation of the NMR signals in vicinity of the phosphorylation sites was detected (Fig. 4)." What is the stoichiometry of Tau phosphorylation by MARK2? What sites are phosphorylated? Do the authors know this from 2D NMR analysis of phospho-Tau?
17. Page 10, Discussion. The protein toxofilin is mentioned; is this from a pathogen? If so, this should be noted and a reference provided.

Response to reviewers for the manuscript:

Multivalent cross-linking of actin filaments and microtubules through the microtubule-associated protein Tau

We thank the referees for their comments and have taken this opportunity to strengthen our manuscript. Please find below a detailed response made in view of the referees' suggestions and concerns.

Reviewer #1: This is an interesting paper that provides the first detailed structural information regarding the interactions of tau with f-actin, and hence is of significance. That said, the paper relies entirely on in vitro observations, which is reasonable for obtaining the structural data, but which handicaps the paper in terms of functional significance and a more general impact. Indeed, despite the number of references to previous work on tau-actin interactions, the physiological significance of these interactions is perhaps not widely appreciated. Data supporting a physiological significance of the observed structural features/interactions would be most appropriate.

Reply: It is true that Tau is perceived by the cell biological and biomedical community mainly as a MT-associated protein, owing to its original discovery as a MAP by Kirschner & colleagues in 1975. However, in the meantime the literature abounds with examples of Tau affecting directly the functions of other proteins by binding to them. This includes not only actin filaments, but also intermediate filaments, RNA-protein particles (e.g. ribosomes, stress granules), proteins involved in motility (e.g. dynactin), membrane-associated proteins (e.g. annexins), signalling molecules (e.g. SH3-module containing proteins, calmodulin) (please see review Mandelkow & Mandelkow, 2012). A search in Pubmed for "actin AND tau" in the title yields 21 hits, but the same search in abstracts already yields 315 publications. To quote two examples from work involving some of the present co-authors, (1) the phosphorylation of Tau via kinases of the MARK and PAK family regulates the relative stability of the microtubule vs. F-actin networks which affects cell migration (Matenia et al., MBC 2005), and (2) the interaction of Tau with the actin cytoskeleton and associated motile machinery is essential for correct neuronal migration in the developing brain (Sapir et al., HMG 2012).

In the present context, a publication of particular relevance is that of Kotani et al. (JBC 1985), building on earlier work of Griffith & Pollard (JBC 1982). They showed that different MAPs (e.g. Tau, MAP2) regulate the higher order assemblies of F-actin filaments, forming either networks (as required for lamellipodia of migrating cells) or bundles (as required for microvilli or microspikes; both types of assemblies are present in postsynapses). Importantly, these assemblies are regulated by posttranslational modifications (e.g. phosphorylation) and additional cofactors (such as calmodulin and calcium). We hope that these examples suffice to document the *in vivo* importance of Tau's effect on the assembly forms of actin, and to shift the perceived view of Tau as a microtubule interactor to Tau's more general functions.

To further support the biological relevance of the Tau-actin interaction, we have added the following statement to the revised version of the manuscript (page 12): *"Moreover the tight connection between microtubules and actin drives the guided extension of axons during neuronal development as well as the formation and activity of synapses in mature neurons (references #65,66). However the molecular mechanisms regulating this cytoskeletal crosstalk are poorly understood. Tau colocalizes with actin in differentiating PC12 cells (reference #67), N2a cells (reference #68) and in the postsynaptic compartments of mature neurons (reference #69). Tau has also been found on dynamic microtubules in the actin rich growth cone areas of developing neurons (references #70,71). These data support a role for Tau as regulator of microtubules and actin and suggests its potential role as linker between the both cytoskeltons."*

Technical issues

1. Binding curves for all data in Fig. S2 should be included.
2. A zoomed in version of the PRE data should be provided that includes only the regions with attenuated signals so the location of the binding sites can be visualized more easily to confirm/assess the described boundaries (L243-A246, K259-K267, V275-L284, S289-S293, S305-V313, S320-H330 and K375-F378).
3. The methods section appears to only include a description of the structural calculation for a single (presumably the 254-290) peptide. Details for the structure calculations for all three peptides should be explicitly included (if they are all the same, this should be indicated).
4. Resonance assignments obtained from the 2D Tocsy experiments for all peptides should be included as a table.

Reply: 1.-4. were implemented in the revised version of the manuscript: Binding curves are included into the new Fig. S1, Fig. S4c shows a zoomed version of the PRE data, resonance assignments are listed in tables S1-S3. Structure calculations were identical for all peptides.

5. The NOE patterns for the 292-319 fragment are quite different from those typically observed for helical structure, which is defined by $i-i+3$ and $i-i+4$ restraints. In addition, the quality of the calculated structures is quite low, with significant populations in the generously allowed and disallowed regions of the Ramachandran plot. In this case, one wonders to what extent the calculated structure actually represents the actual structure. I think a more detailed frank description of the structure calculation results and their quality is required.

Reply: We agree that the number/distribution of restraints are not fully typical for an alpha-helix. We again double-checked all restraints, but could not identify additional restraints in the transfer-NOE spectra. We then analyzed the secondary structure in the ensemble of 20 lowest energy structures (Fig. S12) using the STRIDE server (webclu.bio.wzw.tum.de/stride). STRIDE indicated that there is quite some variation in the location and identity of helical regions across the 20 lowest energy structures. In most structures a 3-10 helix was identified for residues 315-318, although in the lowest energy structure residues 315-318 are in α -helical conformation. The analysis suggests that the transfer-NOE data witness the formation of helical structure in Tau(292-319) upon binding to F-actin, but the number of detected restraints is not sufficient to define a unique conformation. This is described in the revised version of the manuscript on page 9, second paragraph.

6. The data used for the calculation of the isolated helix fragment (254-268) should be included, along with information regarding the structure calculation/validation, as for the other two fragments (as for figures S4 and S5).

Reply: The data have been included into new Figs. S13 and S14.

7. For all calculated structures, an indication should be provided of the criteria used to identify alpha-helical structure, and the boundaries of such structure. The structural ensembles exhibit structure that looks rather atypical for alpha-helix, and this should be clarified. The authors should keep in mind that many non-experts may read their work, providing a vague description of the structural results will result in misinterpretation of the results.

Reply: Thanks for pointing this out. For the revised version of the manuscript, we analyzed the secondary structure in the ensembles of 20 lowest energy structures for the three peptides using the STRIDE server (webclu.bio.wzw.tum.de/stride). For Tau(254-265), STRIDE consistently identified α -helix for residues 259-265. For the lowest energy structure of Tau(254-290) STRIDE detected α -helix for residues 261-268 and 3-10 helix for residues 277-280. In some other conformers of Tau(254-290), however, residues 261-265 were identified as 3-10 helix and residues 277-283 as α -helix. Similar variation in helical conformation was observed for Tau(292-319) (please see above). The analysis suggests that the transfer-NOE data support the formation of helical structure in the three Tau peptides upon binding to F-actin,

but the number of detected restraints is not sufficient to define a unique conformation. This is now clearly described in the revised version of the manuscript (page 7 and page 9). In addition, the boundaries of helical structure (alpha-helix or 3-10 helix) of the lowest energy conformation of each peptide are specified in the text.

8. For all calculated structures, PDB files should be deposited in the protein data bank.

Reply: PDB files were deposited: 5NVB for Tau(254-268), 5N5A for Tau(254-290), 5N5B for Tau(292-319).

9. The true test of the validity and relevance of structural complexes is structure-function studies and these are curiously missing from this work. The conclusions would be substantially strengthened if the structural work was used to design mutations that could perturb the tau-actin interaction and results in quantifiably altered function (at a minimum in bundling assays, much better yet in some in situ or in vivo assay of the significance of tau-actin interactions).

Reply: The structural studies (Fig. 3c,e) suggest that the side-chain of L266 is in direct contact with F-actin. We therefore replaced L266 in Tau(254-290) by a glutamate, followed by F-actin binding studies of Tau(254-290)-L266E and wild-type Tau(254-290) using Saturation Transfer Difference (STD) experiments. The new figure S10 shows that the STD signal of Tau(254-290)-L266E was decreased with respect to the wild-type peptide, demonstrating decreased affinity of the mutant peptide towards F-actin. Moreover, no F-actin specific cross-peaks were observed in the NOESY experiment of Tau(254-290)-L266E in presence of F-actin (Fig. S10 a,b), i.e. only for the wild-type but not for the mutant peptide a transfer-NOE effect was detected. In agreement with a decreased affinity of Tau(254-290)-L266E towards actin, Tau(254-290)-L266E was less efficient in bundling of F-actin when compared to the wild-type peptide (Fig. S10f).

Further support for the validity and relevance of the structural complexes is provided by the results from phosphorylation of KXGS motifs (Fig. 4). S262 is part of the KXGS motif in repeat R1 and located within the interface of the Tau(254-290)/F-actin complex model (Fig. 3c,e). Replacement of S262 by a phosphoserine in Tau(254-290) decreased the peptide's ability to bundle F-actin (Fig. 4f). In addition, phosphorylation of multiple KXGS motifs by MARK2 attenuates binding of Tau to F-actin (Fig. 4d), in particular in vicinity of the phosphorylation sites, and Tau-mediated F-actin bundling (Fig. 4e). We feel that the combined data nicely support the structural findings. Although we agree that cellular and in vivo studies would be interesting, introduction of mutated/phosphorylated Tau will result in multiple effects (at least actin and microtubule binding/bundling, but also other potential effects such as cofilin binding, ...) and would make it difficult to establish a strong connection to our structural work.

10. In the discussion, the authors indicate that toxofilin, another f-actin binding protein, binds in a similar inverted orientation as the 254-290 fragment. It would be informative to include a comparison/overlay with the toxofilin complex as an additional supplementary figure.

Reply: Now shown in Fig. S9.

11. It seems only natural that the bundling ability of MARK2-phosphorylated tau should be assayed and described. I'm surprised this wasn't included.

Reply: We performed a co-sedimentation assay with MARK2-phosphorylated Tau in presence of F-actin. Unlike unphosphorylated Tau, MARK2-phosphorylated tau was unable to bundle actin filaments (Fig. 4e).

12. In the discussion, the authors write ‘Interaction with the proline-rich region is primarily of electrostatic nature’. They should remind the readers that this is based on Fig. S1, as this is only very briefly mentioned in the results, and not explicitly interpreted in this manner.

Reply: A hydrophobicity plot has been added to Fig. S1 and is referenced in the text.

13. The authors propose a competition between microtubules and actin for certain tau binding sites. It would be nice to see this tested/demonstrated experimentally. The authors note that flexible regions connect the different binding sites, but the size of these filamentous assemblies are quite large, and it is unclear to what extent the sterics would allow the proposed cross-linking, especially between near-by sites.

Reply: We proposed the model of a potential competition between actin and microtubules based on the identified binding sites (Fig. 5). In addition, the model is supported by our observation that short Tau peptides with only two actin-binding sites, which are linked by a flexible linker, induce bundling of F-actin (Fig. 3a and Fig. 5b). The later finding indicates that cross-linking of actin filaments by near-by sites is possible (i.e. not precluded by sterics). Please also note that cross-linking of F-actin and microtubules by Tau has been experimentally demonstrated (ref #7). Regarding the experimental demonstration of competition of certain Tau sites for binding to actin and microtubules, we currently do not know how we could best demonstrate it. This is because Tau contains multiple binding sites for F-actin and microtubules, some of them overlapping some others not. The dynamic and promiscuous nature of the interaction of Tau with both microtubules and F-actin complicates competition experiments focused on individual Tau binding sites. To stress that Fig. 5c is only a model, we state in the revised version of the manuscript “*Thus, the availability of multiple Tau sites for binding to microtubules and F-actin together with Tau residues, which are dedicated for binding to only F-actin or microtubules (Fig. 5b), suggests a model in which microtubules and actin filaments are cross-linked by Tau in a multivalent, dynamic manner (Fig. 5c), in agreement with experimentally observed cross-linking of microtubules and actin filaments by Tau (ref #7)*”.

Responses to reviewer 2

Reviewer #2: On the whole, this is an impressive study that points to both similarities and differences in the mechanisms by which tau binds to F-actin versus microtubules. It should be noted, however, that the physiological significance of tau interactions with actin filaments *in vivo*, including the extent, if any, to which such interactions occur in normal and diseased states, is not well established. With that caveat in mind, the crucial involvement of tau in the pathogenesis of tauopathies, including Alzheimer’s disease, indicates that this study is bound to attract the attention of scientists who are interested in macromolecular structures germane to neurodegeneration.

Reply: Thanks for the supportive words. Regarding the *in vivo* relevance of the Tau-actin interaction, we have added more references and additional text to the discussion section (page 12). In addition, please see also our comment to reviewer 1. We believe that the provided examples document the *in vivo* importance of Tau's effect on the assembly forms of actin, and also highlight the more general (beyond MT interaction) functions of Tau.

Reviewer #2: One issue that deserves further attention concerns Figure 5C, which shows how tau can hypothetically cross-link actin filaments to microtubules. This figure should be supported by experimental evidence, or the figure and all discussion of it should be removed.

Reply: Please note that cross-linking of F-actin and microtubules by Tau has been experimentally demonstrated (ref #7). We therefore feel that it is important to put our results on the Tau-actin interaction into the context of Tau-MT interaction and MT-actin crosslinking (ref #7). Obtaining high-resolution insight into a complex of Tau bound simultaneously to both MT and actin filaments is highly challenging because of the heterogeneous and promiscuous nature of the system. We therefore feel that Fig. 5c helps understanding the complex nature of Tau-based actin-MT cross-linking and hopefully motivates further experiments. In order to avoid any misinterpretation of Fig. 5c, however, we now clearly stress in the revised version of the manuscript that Fig. 5c is only a model (page 12): *“Thus, the availability of multiple Tau sites for binding to microtubules and F-actin together with Tau residues, which are dedicated for binding to only F-actin or microtubules (Fig. 5b), suggests a model in which microtubules and actin filaments are cross-linked by Tau in a multivalent, dynamic manner (Fig. 5c), in agreement with experimentally observed cross-linking of microtubules and actin filaments by Tau (ref #7)”*.

Minor points:

- 1) It would be helpful to readers if all figures were so self-explanatory that shifting attention away from the figures in order to read the figure legends would be unnecessary. To that end, please directly label the positions of tau and actin on Figure 1c; of tau, actin and cofilin on Figure 2c; and of actin on Figure 4c. Also, please indicate the lengths of the scale bars directly on Figures 1b and 3a.
- 2) The NBD-actin fluorescence change assay is not widely known and is not referenced. Please add the original reference for this technique.
- 3) The following sentence on page 4 is somewhat misleading: “After addition of cofilin, Tau’s NMR resonance intensities and positions were similar to those in the absence of F-actin (Fig. 2b), indicating that (i) Tau is no longer bound to F-actin and (ii) cofilin does not bind to Tau.”. While the statement is consistent with the data shown, it neglects to consider the possibility that the cofilin severed the actin filaments, which then partly depolymerized, which in turn would lead to decreased tau binding to F-actin.
- 4) Figure S3 has 3 parts: (a), (b) and (c), but part (b) is labeled as (a).
- 5) The following statement from the bottom of page 8 to the top of page 9 seems incorrect: “in agreement with the sequence similarity of Tau(292-319) (corresponding to part of R1/R2) with Tau(254-290) (corresponding to part of R2/R3)”. The sentence should state that Tau(292-319) corresponds to R2/R3, while Tau(254-290) corresponds to R1/R2.
- 6) On page 9, the term “wild type” is awkwardly used to refer to wild type tau that had not been phosphorylated in vitro by MARK2. Please change “wild type” to “non-phosphorylated”.

Reply: Thanks for spotting. We implemented all changes in the revised version of the manuscript.

Responses to reviewer 3

Reviewer #3: Overall, the manuscript is well written and the NMR and other data are of high quality. This reviewer accepts the authors’ general conclusions but would like to see the sequence-specificity of Tau motif/actin interactions tested through mutagenesis.

Reply: Thanks for the supportive words. The structural studies (Fig. 3c,e) suggest that the side-chain of L266 is in direct contact with F-actin. We therefore replaced L266 in Tau(254-290) by a glutamate, followed by F-actin binding studies of Tau(254-290)-L266E and wild-type Tau(254-290) using Saturation Transfer Difference (STD) experiments. The new figures Fig. S10c-e show that the STD signal of Tau(254-290)-L266E was decreased with respect to the wild-type peptide, demonstrating decreased affinity of the

mutant peptide towards F-actin. Moreover, no F-actin specific cross-peaks were observed in the NOESY experiment of Tau(254-290)-L266E in presence of F-actin (Fig. S10a,b), i.e. only for the wild-type but not for the mutant peptide a transfer-NOE effect was detected. In agreement with a decreased affinity of Tau(254-290)-L266E towards actin, Tau(254-290)-L266E was less efficient in bundling of F-actin when compared to the wild-type peptide (Fig. S10f).

Further sequence-specific support for the Tau motif/actin interactions is provided by the results from phosphorylation of KXGS motifs (Fig. 4). S262 is part of the KXGS motif in repeat R1 and located within the interface of the Tau(254-290)/F-actin complex model (Fig. 3c,e). Replacement of S262 by a phosphoserine in Tau(254-290) decreased the peptide's ability to bundle F-actin (Fig. 4f). In addition, phosphorylation of multiple KXGS motifs by MARK2 attenuates binding of Tau to F-actin (Fig. 4d), in particular in vicinity of the phosphorylation sites, and Tau-mediated F-actin bundling (Fig. 4e).

Reviewer #3: Also, it would be nice to have interaction mapping data from at least a second MTSL-labeled form of actin to complement the single dataset that utilizes Cys 374.

Reply: Please note that the actin employed in our study is purified from natural source (as commercially available from the company Cytoskeleton). We could now try to express human actin recombinantly, which however would have to be done in insect cells (e.g. SF9 cells; please see Meola et al. *J Struct Biol.* 2014 Oct;188(1):71-8). Because my lab has no experience with expression of proteins in insect cells and this usually requires a good amount of experience, it is not clear how long this endeavor would take and if it would be successful at all. Alternatively, we might try to purify actin from yeast (Feng et al. *J Biol Chem.* 1997 272(27):16829-37), however, then we would no longer work with human actin with unknown consequences on the interaction with Tau. We could thus only try to collaborate with a lab, which has successfully prepared human actin in insect cells. This, however, would require that this lab does not have a conflict of interest. Given that we have additional data (in particular competition with cofilin in both NMR and F-actin bundling assays; please see Figs. 2b,c) that support binding of Tau to the hydrophobic pocket of actin, we hope that the referee agrees that inclusion of data with recombinant actin is beyond the scope of the current manuscript.

Technical issues

1. Suppl Fig 1 does not address hydrophobicity, as noted on page 4, line 105. The authors should include an analysis of hydrophobicity for Tau in order to discuss this issue.
2. Switching between I/I0 and (1-I/I0) formats for presenting NMR binding site mapping data is confusing. It is suggested that a single format should be used throughout the manuscript.
3. Control lanes are lacking in Fig. 2c showing migration of the three individual components in the experiment. These control lanes need to be included in the gel.

Reply: 1.-3. were implemented in the revised version of the manuscript.

4. The authors should perform a titration of F-actin into ¹⁵N-Tau to determine local K_d values for residues within the various binding motifs. Do these values reflect the value from FP? This relates to comment #6, below.

Reply: We performed the titration as suggested. The figure below illustrates the data for three selected residues (I260, L266, G287). For all three residues, the grid-search in numerical solution of the (two-state) Bloch-McConnell equation was pushed to the edge of the parameter space: K_d of 10 μM, k_{on} of 10⁷ M⁻¹s⁻¹ and R₂,bound of 1000 s⁻¹. Still, fits were not good. Based on the profile, we think that the interaction of Tau with F-actin may follow at least a three-state system: Tau binds F-actin, then undergoes a conformational transition there. This process might be simulated by a three-state BM

equation, however, it would involve many variables. In addition, a further complication might be that the interaction of Tau with F-actin is not monophasic, but similar to the Tau-MT interaction biphasic (Ackmann et al. JBC 2000). It is therefore difficult to determine local K_d values. Please see also our reply to comment #6.

5. The authors state that only one of five Cys residues (C374) is labeled with MTSL. Is this known experimentally? Certainly Cys 374 is highly accessible to solvent. The authors should compute the solvent accessibility of the other five to support their argument that they are inaccessible. How was MTSL labeling confirmed? What was the stoichiometry of labeling? Mass spectrometry could be used to address this issue. Some type of analytical data need to be provided to document MTSL labeling at Cys 374.

Reply: We have included into the revised Supplementary Fig. 4 a table with the solvent accessibilities of the five Cys residues. Already in the 1970s/1980s several EPR studies with Cys374 spin-labelled actin have been published (e.g. references #35-39 in the revised manuscript). From these studies it was concluded that Cys374 is most solvent exposed and incorporates more than 70% of spin labels during spin labeling reactions (references #35-39). This is now described on page 5 (2nd paragraph) of the revised manuscript. In addition, the authors of reference #39 report that actin in which Cys374 was blocked by MalNet could not be labeled with N-ethylmaleimide. Thus, although we cannot exclude that other cysteine residues besides Cys374 might be labeled by MTSL, this should be at best at low levels and thus does not strongly influence the interpretation of the PRE data. Please note that the competition experiments with cofilin (Fig. 2b,c) provide further support for binding to the hydrophobic pocket of actin.

6. Figs. 1e & f, 2b & e. The authors state that the resonances of 15N-Tau are broadened beyond detection upon binding to F-actin polymers; consequently, peaks do not shift upon binding but rather disappear. With a K_d value of 60 nM (from FP), the fraction of free Tau under the conditions of the NMR experiments would be smaller than 20% but 20% to 40% of the starting signal persists in the presence of F-actin according to these figures. What is the explanation for the residual NMR signals? The authors should perform a titration of F-actin into 15N-Tau to determine residue-specific K_d values and determine whether they agree with the FP-derived value. The different actin binding motifs of Tau probably behave independently and may exhibit weak local K_d values but may function collectively to give the observed overall tight binding—a type of dynamic complex. The authors should clarify the

situation and provide a more quantitative description of this binding event based on NMR data.

Reply: Please note that the FP and NMR measurements were done with different buffer conditions. The FP experiment was performed at lower ionic strength (GAB; 5mM TrisHCl pH 8.0, 0.2mM CaCl₂, 0.2 mM ATP, 0.5mM DTT) when compared to the NMR measurements (50mM NaH₂PO₄ / Na₂HPO₄ pH 6.8, 10mM NaCl, 1mM DTT), in which the assignments of Tau are available (Mukrasch et al., PLOS Biol 2009). In the revised version of the manuscript, we now also report NMR spectra at low ionic strength (5mM TrisHCl pH 6.8, 0.2mM CaCl₂, 0.2 mM ATP, 0.5mM DTT), which demonstrate complete disappearance of the NMR signals of the Tau residues that bind to F-actin (Fig. S2b,c), in agreement with FP-derived affinities (Fig. S1). Regarding the determination of local K_d values, please see our reply above.

7. Fig. 1e. The authors should show the full 2D HSQC spectrum as a supplemental figure.

8. Page 5, Line 148. "Notably, no PRE effect was observed at the C-terminus." What residues constitute the "C-terminus"? There are PRE effects between residues 380-400. Please clarify and revise, if needed.

9. Pages 5-6. Please define the residue boundaries of the Tau K18 construct.

10. Page 6. What is the binding constant of the residue 254-290 Tau peptide for F-actin? This should be reported.

11. Fig. 3b. Please show zoom views of key regions of the 2D NOESY spectrum as Supplemental figures, including the full amide region and the H_α-HN region. Also, please display these with symmetric F1 & F2 dimension limits/scales, as is the usual convention.

Reply: 7.-11. were implemented in the revised version of the manuscript (please see figures S1, S2, S5, S7, S11, S13).

12. Page 7, top. "In contrast, NOE cross-peaks, which are specific for α-helical conformation, were observed for residues 259-267 and residues 275-284. The corresponding residues in full-length Tau experienced pronounced PRE broadening (Fig. 2e), in agreement with binding to F-actin's hydrophobic pocket." The PRE profile in Fig. 2e appears to extend beyond residue 284 as far as almost to residue 300. Why did the authors not investigate a slightly longer Tau peptide?

Reply: The PRE effect decreases strongly at about residue 293, but the diamagnetic intensity ratio $I_{\text{bound}}/I_{\text{free}}$ increases already earlier (Supplementary Fig. 4c). Because of the long-range nature of the PRE effect, PRE broadening is expected also for residues, which are in close proximity to the binding site. Because medium-range NOEs did not extend beyond residue 288 (Supplementary Fig. 6b), extension of the peptide is not expected to have a strong effect.

13. Fig. 3f. It is difficult to understand how the inverted Tau 254-268 region could bind to F-actin the same way as cofilin in the noted PDB file because the 110-127 region of cofilin forms a continuous helix while Tau has a three residue long gap in the sequence alignment. Can the authors please comment on this and revise the manuscript, if needed.

Reply: The superposition of the structures (Fig. 3e) suggests that the 3-residue gap in sequence (Fig. 3f) is compensated by a more extended structure in F-actin bound Tau, when compared to the α-helical conformation observed for residues S119-A123 of F-actin bound cofilin. This is now stated on page 8 of the revised manuscript.

14. Suppl. Table S1. The authors should include all of the Haddock clusters and their respective scores/statistics. What restraints were used in the docking experiments? A restraints violation energy is given. How have the authors validated their structural model? They should perform mutagenesis of residues in Tau that appear to be critical for binding to F-actin and i) demonstrate loss of binding by NMR, and ii) altered docking results using Haddock.

Reply: The Haddock clusters and their scores/statistics have been included into Supplementary Fig. 8. The docking model (Fig. 3e) indicates that the side-chain of L266 makes a hydrophobic contact with F-actin. We therefore prepared a mutant version of Tau(254-290), in which L266 was replaced by a

glutamate, i.e. Tau(254-290)-L266E. We then used saturation transfer difference (STD) experiments to probe the influence of the mutation on binding of Tau(254-290) to F-actin. When compared to the wild-type peptide, Tau(254-290)-L266E showed a decreased STD signal (Fig. S10c-e), indicating that the mutation attenuated F-actin binding. Moreover, no additional cross-peaks were observed in the NOESY spectrum of Tau(254-290)-L266E in the presence of F-actin (Supplementary Fig. 10a,b), i.e. no transfer NOE effect occurred. In agreement with a decrease in affinity of Tau(254-290)-L266E for binding to F-actin, the mutant peptide was less efficient in promoting F-actin bundling (Supplementary Fig. 10f).

Further sequence-specific support for the Tau motif/actin interactions is provided by the results from phosphorylation of KXGS motifs (Fig. 5). S262 is part of the KXGS motif in repeat R1 and located within the interface of the Tau(254-290)/F-actin complex model (Fig. 3c,e). Replacement of S262 by a phosphoserine in Tau(254-290) decreased the peptide's ability to bundle F-actin (Fig. 4f). In addition, phosphorylation of multiple KXGS motifs by MARK2 attenuates binding of Tau to F-actin (Fig. 4d), in particular in vicinity of the phosphorylation sites, and Tau-mediated F-actin bundling (Fig. 4e).

Please note that Haddock is not able to predict the effect of mutations.

15. Suppl. Fig. 5. The authors should illustrate regions of the 2D NOESY spectra for the Tau(292-319) peptide from which structural restraints were extracted (as discussed in comment # 11, above).

Reply: We have included the requested figures into the new Suppl. Fig. 11.

16. Page 9. "We phosphorylated full-length Tau by MARK2, followed by NMR measurements in the absence and presence of F-actin. In contrast to the wild-type protein, little perturbation of the NMR signals in vicinity of the phosphorylation sites was detected (Fig. 4)." What is the stoichiometry of Tau phosphorylation by MARK2? What sites are phosphorylated? Do the authors know this from 2D NMR analysis of phospho-Tau?

Reply: Sites and stoichiometries of Tau phosphorylation by MARK2 have been described in detail in our previous publication (reference #53). In the revised version of the manuscript we have added an additional figure (Fig. 4a) and state in the manuscript: "*The downfield chemical shift of phosphorylated residues (Fig. 4a) is in agreement with previous reports and confirms phosphorylation at S262, S293, S305, S324, S356 and S416 (reference #53)*".

17. Page 10, Discussion. The protein toxofilin is mentioned; is this from a pathogen? If so, this should be noted and a reference provided.

Reply: The source of toxofilin is now mentioned in the legend to Supplementary Fig. 9.

REVIEWERS' COMMENTS:

Reviewer #1 (Remarks to the Author):

The revised manuscript is considerably improved and addresses many of the concerns that were raised in the initial review. An improved description of the potential relevance of tau-actin interactions, including additional references, is provided, establishing more convincingly the physiological relevance of the work, although no studies in situ or in vivo have been added. The requested structure determination spectra, parameters, protocols, and results have been provided, and the protocol for identification of helical structure has been clearly defined. The departure of the determined structures from classical helical geometry is now satisfactorily noted and described. Analysis of a mutant based on the structural model is also included and shows that a perturbation at the putative binding interface reduce F-actin binding by tau. The effect of MARK2 phosphorylation of tau on F-actin bundling is also characterized.

A few concerns remain that should be addressed.

Panels a and b of Supp. Fig. 1 are not consistent with each other. The binding curve for tau254-290 shows the weakest binding in panel 1, but the Kd is shown as lower than 254-284 and 254-282s262p. It's not clear why 254-290 would bind more weakly than 254-284 and some comment/explanation should be provided.

The authors have added a single mutant that reduces STD and reduces actin bundling. This is nice, but NBD-based F-actin binding data for this mutant should be added to Supp. Fig. 1 to quantify the apparently decreased affinity and provide a direct comparison to the other constructs.

At the end of the results section, the authors write 'Phosphorylation within one actin-binding motif of Tau has however little influence on the Tau/F-actin interaction at the other six sites, providing support for multiple independent F-actin binding sites in Tau.' This statement is not directly supported by any data. To show this, the authors would have to generate singly-phosphorylated full length tau (or k18) and show that actin binding is only perturbed at this one phosphorylation site. Thus, this statement should be removed or qualified.

Reviewer #3 (Remarks to the Author):

The authors have appropriately addressed this reviewer's comments on the original manuscript; the revised manuscript is suitable for publication.

Response to reviewer #1 for the manuscript:

Multivalent cross-linking of actin filaments and microtubules through the microtubule-associated protein Tau

Reviewer #1: Panels a and b of Supp. Fig. 1 are not consistent with each other. The binding curve for tau254-290 shows the weakest binding in panel 1, but the K_d is shown as lower than 254-284 and 254-282s262p. It's not clear why 254-290 would bind more weakly than 254-284 and some comment/explanation should be provided.

Reply: Thanks for pointing this out. The different slopes in panel a are due to different concentrations of F-actin/variations in F-actin preparation, which were used for some of the peptides. Because experimental F-actin concentrations depend on the efficiency of polymerization, F-actin concentrations were a fit parameter when deriving K_d values. The respective values are now mentioned in the figure legend.

Reviewer #1: The authors have added a single mutant that reduces STD and reduces actin bundling. This is nice, but NBD-based F-actin binding data for this mutant should be added to Supp. Fig. 1 to quantify the apparently decreased affinity and provide a direct comparison to the other constructs.

Reply: The data have been included into Supp Fig. 1.

Reviewer #1: At the end of the results section, the authors write 'Phosphorylation within one actin-binding motif of Tau has however little influence on the Tau/F-actin interaction at the other six sites, providing support for multiple independent F-actin binding sites in Tau.' This statement is not directly supported by any data. To show this, the authors would have to generate singly-phosphorylated full length tau (or k18) and show that actin binding is only perturbed at this one phosphorylation site. Thus, this statement should be removed or qualified.

Reply: The statement has been removed.